psychology

self-paced preparation, proactive cognitive control, instruction-based learning, novel instructions, cost–benefit analysis

**Author for correspondence:**
Christina B. Reimer
e-mail: reimerc7@gmail.com

# Benefits and costs of self-paced preparation of novel task instructions

Christina B. Reimer, Zhang Chen and Frederick Verbruggen

Department of Experimental Psychology, Ghent University, Ghent, Belgium

 CBR, 0000-0003-2496-3971; ZC, 0000-0002-3500-9182; FV, 0000-0002-7958-0719

Rapidly executing novel instructions is a critical ability. However, it remains unclear whether longer preparation of novel instructions improves performance, and if so, whether this link is modulated by performance benefits and costs of preparation. Regarding the first question, we reanalysed previous data on novel instruction implementation and ran Experiment 1. Experiment 1 consisted of multiple mini-blocks, in which participants prepared four novel stimulus–response (S-R) mappings in a self-paced instruction phase. After participants indicated they were ready, one of the four stimuli was presented and they responded. The reanalysis and Experiment 1 showed that longer preparation indeed led to better performance. To examine if preparation was modulated when the benefits of preparation were reduced, we presented the correct response with the stimulus on some trials in Experiments 2 and 3. Preparation was shorter when the probability that the correct response was presented with the stimulus increased. In Experiment 4, we manipulated the costs of preparation by changing the S-R mappings between the instruction and execution phases on some trials. This had only limited effects on preparation time. In conclusion, self-paced preparation of novel instructions comes with performance benefits and costs, and participants adjust their preparation strategy to the task context.

## 1. Introduction

The ability to rapidly execute novel instructions greatly facilitates flexible and purposeful behaviour in everyday life, for example when trying a new recipe for dinner, when using a new piece of equipment, or when assembling a new piece of furniture. Recently, the implementation of instructions in the human brain and the effects of such instructions on performance have gained renewed interest [1]. In the present study, we focused on self-paced preparation of novel instructions, and in particular, how

the duration of self-paced preparation (a proxy for more careful processing of instructions) is modulated by context (costs and benefits). But before we could address this main question, we had to establish the basic relationship between self-paced preparation time and performance (Study Aim 1). One would assume that people perform better (i.e. fewer errors, faster responses) after preparing the to-be-performed task for a longer period. Surprisingly, this intuition has been challenged, as evidence for a link between self-paced preparation time and performance is still mixed [2,3]. To preview our own findings, we found that prolonged self-paced preparation did indeed lead to a performance benefit, both in the careful reanalysis of a previous study and in a new experiment. Encouraged by these findings, we then investigated in a second series of experiments whether this positive link was modulated by expected costs and benefits of self-paced preparation (Study Aim 2). We will introduce the previous work on costs and benefits of self-paced preparation after discussing the reanalysis and the first experiment.

## 2. Aim 1: Does prolonged self-paced preparation lead to a performance benefit?

In recent years, the ability to rapidly implement new instructions gained renewed interest [3–7]. Most of these studies focused either on the neural mechanisms or the behavioural consequences of the (successful) implementation of such instructions (as discussed in more detail below). Often, self-paced preparation was used in these behavioural experiments. The duration of this interval was rarely a dependent variable of interest though, and the few studies that have focused on it have produced inconsistent results.

Based on research in other domains (and intuition), one would predict a strong and positive link between the duration of self-paced preparation and task performance, under the assumption that careful preparation requires more time (akin to the speed–accuracy trade-off in decision-making [8]). But evidence is still mixed. In a seminal study, Meiran and colleagues observed a *negative* link between self-paced preparation time and performance [2]. In their task-switching study (Experiments 1 and 2), participants were asked at the beginning of each trial to indicate when they were ready (i.e. fully prepared) to start the next trial. Surprisingly, in both experiments, an increase in self-paced preparation time (i.e. prepRT) was associated with an increase in reaction time (i.e. RT), and thus poorer performance. This RT increase across bins was not clearly associated with an increase in accuracy (ACC). This finding led them to conclude that participants were only poorly aware of their state of preparation with respect to performance on the upcoming practised task, and guessed their level of preparedness based on indirect information (e.g. number of mappings or tasks that had to be prepared). Longman *et al.* similarly found longer RT with longer self-paced preparation (without an ACC improvement) in their task-switching study with practised tasks [9]. In this study, longer self-paced preparation led to a reduction in switch cost (i.e. the difference between task repetitions and task switches), but this effect was primarily driven by an increase of RT on task-repetition trials rather than a decrease on task-switch trials. Combined, the two studies suggest that self-paced preparation does not necessarily lead to better performance, at least within a context in which people have to switch frequently between well-practised tasks.

The studies that (unexpectedly) found a negative link focused on the execution of well-practised tasks. The pattern may look different though when people prepare novel tasks. In a study by Cole *et al.* [3], participants prepared both novel and practised tasks in an experimenter-paced and a self-paced condition (Experiment 1). After experimenter-paced preparation, ACC was higher in practised tasks compared to novel tasks. However, after self-paced preparation, ACC was similar in novel and practised tasks. The authors replicated this finding in another experiment, in which all trials were self-paced (Experiment 3). Furthermore, two regression analyses revealed that the link between longer self-paced preparation and higher accuracy was stronger for novel tasks than for practised tasks. First, using a single-trial regression, the authors found that participants had higher ACC in novel tasks when they prepared for a longer period compared to when they prepared for a shorter period. For practised tasks, the authors observed a non-significant negative relationship (i.e. ACC numerically dropped when preparation time increased). Second, an individual-differences analysis (using the data of Experiment 1 and 3) showed that participants who prepared the task instructions longer were more accurate; this effect was stronger for novel tasks than for practised tasks. Combined, these two analyses suggest a positive link between self-paced preparation and task performance.

The inconsistency between previous studies (i.e. positive versus negative links between self-paced preparation and task performance) could be simply due to the use of different procedures (i.e. self-

paced preparation of novel tasks versus well-practised tasks). But given the low number of relevant studies, we deemed it crucial to independently establish the link first. We did this in two steps. First, we reanalysed the data of a cross-sectional developmental study that focused on how different age groups implemented novel task instructions. Encouraged by the findings of this reanalysis, we then further developed a novel paradigm specifically for studying the self-paced preparation phase.

# 3. Reanalysis of Verbruggen *et al.* [10]

To establish if there is indeed a positive link between the duration of the self-paced preparation interval and performance in novel tasks, we reanalysed the data of Verbruggen *et al.* [10]. This cross-sectional developmental study examined how different age groups (young children, older children and late adolescents; overall age range: 4–19 years) implemented novel task instructions using a variant of the NEXT paradigm [7,11]. Specifically, the experiment consisted of a series of mini-blocks. At the beginning of each mini-block (the instruction phase), instructions for two novel stimulus–response (S-R) mappings were presented (e.g. 'lion' = left, 'clown' = right; see figure 1, reproduced from Verbruggen *et al.* [10] under CC-BY license). The duration of this phase was self-paced, and participants were told that they had to apply these instructions in the GO phase of the mini-block (which consisted of two trials). Before participants could do so, they had to advance through a NEXT phase (which consisted of up to three trials). In this NEXT phase, stimuli were presented but their identity could be ignored. Participants simply had to press the same NEXT key on each trial (which was either the left or right response key). Even though the S-R mappings (of the GO phase) had never been applied before, participants of all age groups (including the youngest children) responded more slowly to NEXT stimuli when the NEXT response and the GO response were incompatible (the clown requiring a left response in the NEXT phase but a right response in the GO phase) compared to when they were compatible (the lion requiring a left response in both phases). This instruction-based interference effect shows that instructions enable 'automatic' task performance. Furthermore, the finding that the interference effect was present (and even largest) in 4–5-year-old children indicates that even this age group can implement instructions in advance.

In the study of Verbruggen *et al.* [10], the main dependent variable of interest was the interference effect during the NEXT phase. Performance in the GO phase was also analysed as a function of age group, but not as a function of the duration of the self-paced preparation phase. Therefore, we reanalysed the GO data of the original study.

## 3.1. Methods, procedure and reanalysis

We refer readers to the original study for full details about the procedure. In short, 178 children (4–11 years old) and 30 late adolescents (17–19 years old) participated in this experiment. As in the original study, 12 children were excluded (due to poor performance or incomplete datasets). All age groups performed the same NEXT paradigm, which is summarized in figure 1 (reproduced from [10]). This NEXT paradigm consisted of 48 experimental mini-blocks. Each mini-block consisted of a self-paced instruction phase, a NEXT phase (consisting of zero, one, two, or three trials) and a GO phase (always consisting of two trials). At the end of the mini-block, feedback on the GO performance was presented. In the self-paced instruction phase, the instructions remained on screen until participants had pressed a key *and* at least 3 s had elapsed. If they pressed the start key before 3 s had elapsed, the response was registered (explaining why prepRT was sometimes shorter than 3 s), but they had to wait a bit before the NEXT phase started. This was done to ensure that individuals of all age groups would process the instructions to some extent. Note that if participants pressed the start key after 3 s, the NEXT phase would start immediately.

Our analyses focused on self-paced preparation time (i.e. the interval between the presentation of the instructions and the moment participants indicated they were ready; prepRT) and GO performance on the first trial (i.e. the latency and accuracy; GO RT and GO ACC, respectively). Initial explorations of the self-paced preparation time showed that prepRT (and response latencies in general) decreased substantially throughout the experiment. This could mask or confound effects of preparation though (see appendix A for a graphic illustration of the problem). Therefore, we performed a *pair-split* analysis to identify mini-blocks with 'short' and 'long' self-paced preparation. Specifically, for each participant, mini-blocks were arranged in ascending order based on the block number, so that two consecutive mini-blocks formed a pair (i.e. mini-blocks 1 & 2, 3 & 4, 5 & 6, …, 47 & 48). For each pair, prepRTs of the two mini-blocks were then compared with each other to identify the mini-block with 'long' and with 'short' self-paced preparation. Note that when one of the mini-blocks of the pair was

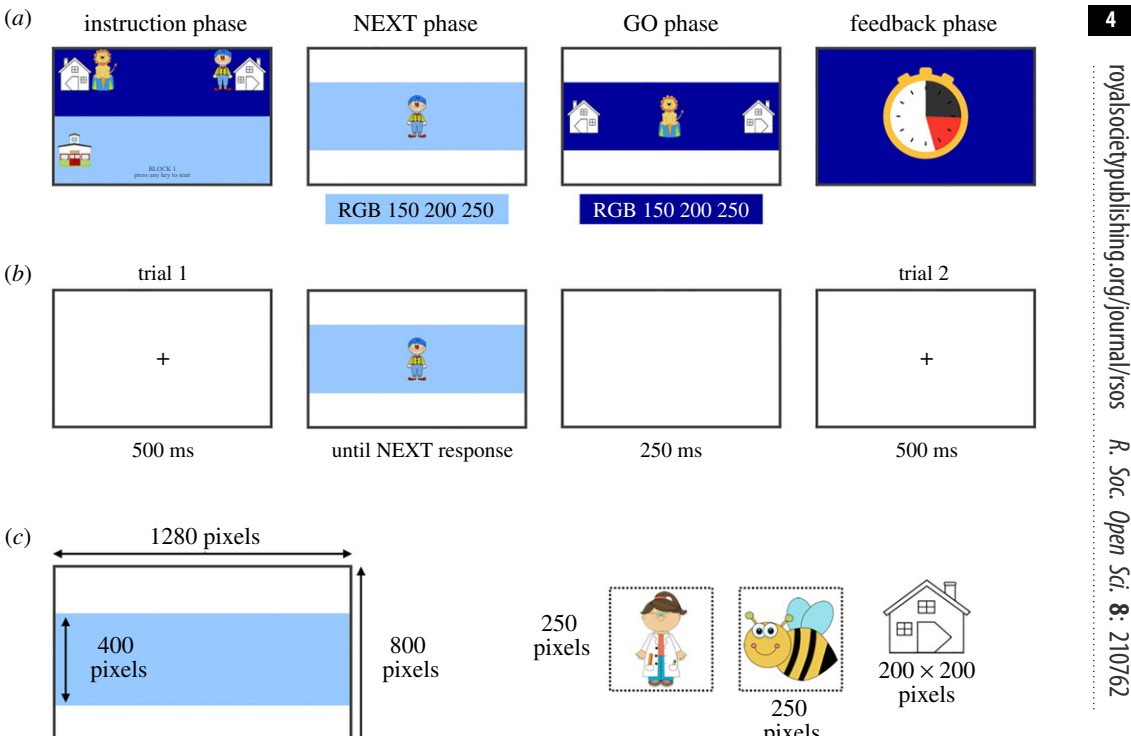

**Figure 1.** A depiction of the course of a mini-block, with four different phases (*a*), and the trial course for NEXT trials (*b*). The trial course for GO trials was very similar to the course for NEXT trials, except that the stimulus disappeared as soon as a response key (correct or incorrect) was pressed. Panel (*c*) shows the size of the screen and the stimuli. Reproduced from [10] under CC-BY license.

excluded (see next paragraph), then the other mini-block of the pair was automatically excluded as well. As illustrated in more detail in appendix A, this pair-split analysis is essentially a median split that controls for global fluctuations, and is therefore preferred over the single-trial regression approach used by Cole *et al.* [3]. (Note that the individual-differences regression approach used by Cole *et al.* is not confounded by global fluctuations either; however, we were exclusively interested in within-subjects effects in the present study, which is why we used our pair-split analysis.)

Based on the pair-split analysis, there were approximately 24 trials per cell and in total (i.e. across all individuals and age groups) 4869 mini-blocks with 'short' and 4869 with 'long' self-paced preparation. Prior to the calculation of relevant means, we excluded mini-blocks in which prepRT were less than 100 ms or greater than 10 000 ms and mini-blocks in which GO RT were less than 100 ms or greater than 10 000 ms (9.98% of trials). The same outlier criteria were used in the original study. In the analyses of GO RT, incorrect trials were further excluded (14.33% of trials).

We then analysed performance on the first GO trial depending on the duration of self-paced preparation (i.e. 'short' versus 'long'). For this, we conducted a paired-samples *t*-test and the Bayesian equivalent (using a Cauchy distribution of width 0.707 as the prior). We reported the *p*-value, the Bayes Factor $BF_{10}$ as well as the effect sizes Hedges's average *g* and Cohen's $d_z$ [12].

For all experiments reported in this manuscript, all data processing and analyses were completed with R (v. 3.6.0 [13]) and RStudio (v. 1.2.1335 [14]) using the tidyverse package (v. 1.3.0 [15]) and the BayesFactor package (v. 0.9.2+ with its default prior [16]). We followed the recommendation of Wagenmakers *et al.* [17] to interpret the Bayes Factors. Experimental scripts and materials, raw data files and data analysis scripts that were used in the present study can be found on the Open Science Framework (OSF, see https://osf.io/vzxfy/).

## 3.2. Results and discussion

Mean prepRT, mean ACC and mean GO RT of the pair-split analysis are shown in figure 2. Inferential statistics of overall task performance (across all age groups) as well as performance for each age group separately (using the same groups as in the original study) are presented in table 1.

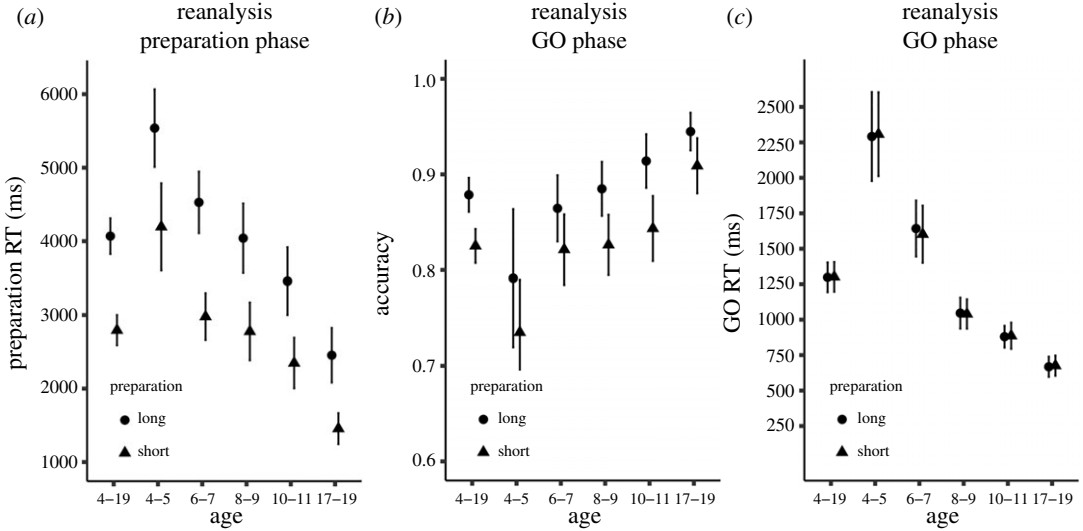

**Figure 2.** Reanalysis of Verbruggen *et al.* [10]. (*a*) Mean preparation reaction time ( prepRT), (*b*) mean accuracy (ACC) and (*c*) mean RT of the first GO trial (GO RT) depending on long and short self-paced preparation trials. Error bars stand for 95% within-subject confidence intervals.

Overall, prepRT was 4069 ms for long self-paced preparation and 2794 ms for short self-paced preparation (obviously, we expect—by definition—a prepRT difference between long and short trials). Importantly, ACC was higher after long self-paced preparation (mean ($M$) = 0.88) than after short self-paced preparation ($M = 0.83$, $BF_{10} = 3\,573\,348$, $p = 0.001$). However, there was no difference between GO RT after long self-paced preparation ($M = 1299$ ms) and after short self-paced preparation ($M = 1302$ ms, $p = 0.78$), and the analysis revealed 'strong' evidence for the null effect ($BF_{10} = 0.08$). Taken together, for all participants, the pair-split analysis provided evidence for a link between self-paced preparation and ACC, but not between self-paced preparation and GO RT.

When looking at the five age groups separately (i.e. 4–5-, 6–7-, 8–9-, 10–11- and 17–19-year-old participants), prepRT differences between short and long self-paced preparation were in the range of 999–1549 ms. ACC was numerically higher after long self-paced preparation than after short self-paced preparation for all age groups (mean differences in ACC between long and short self-paced preparation = 0.04–0.07). However, the statistical evidence for this link varied between 'no evidence' and 'extreme' evidence ($BF_{10}s = 0.64–765$, $ps = 0.001 – 0.11$). There were no consistent GO RT differences between long and short self-paced preparation across the five age groups ($M = -17–40$ ms, $BF_{10}s = 0.15–0.28$, $ps = 0.27–0.9$).

In sum, the reanalysis of Verbruggen *et al.* [10] provided evidence for a positive link between self-paced preparation of novel instructions and performance. Across participants of all age groups, ACC was (numerically) higher after long self-paced preparation than after short self-paced preparation. It should be noted though that the statistical analysis revealed that this effect was only robust from age group 8–9 onward. GO RT, however, was similar after short and long self-paced preparation. The findings are consistent with the findings of Cole *et al.* [3], even after controlling for global fluctuations. They indicate that—at least for self-paced preparation of novel tasks—longer preparation latencies are associated with more accurate task performance. In other words, we observed a speed–accuracy trade-off at the task preparation level.

In the NEXT paradigm, participants implemented novel task instructions in a self-paced manner. Research on instruction-based learning has shown that the prefrontal cortex is strongly engaged in the implementation process [1,5,18]. Given that the prefrontal cortex is still developing during early childhood and adolescence as compared to adulthood, it is remarkable that the link between self-paced preparation of novel instructions and performance in the GO phase was virtually present across all age groups in the reanalysis. Thus, the ability to proactively prepare for novel tasks is already present during early childhood and develops further during adolescence until adulthood.

## 4. Experiment 1

The reanalysis of Verbruggen *et al.* [10] indicates that longer self-paced preparation of novel tasks is associated with higher accuracy in the execution phase. In this previous study, the NEXT paradigm

**Table 1.** Statistical results of the reanalysis of Verbruggen et al. [10].

| age | measure | diff | lowerCI | upperCI | t | p | $BF_{10}$ | $g_{av}$ | $d_z$ |
|---|---|---|---|---|---|---|---|---|---|
| overall | PrepRT | 1288.91 | 1204.11 | 1373.72 | 29.97 | <0.001 | $2.69 \times 10^{72}$ | 0.8 | 2.11 |
|  | ACC | 0.05 | 0.04 | 0.07 | 6.25 | <0.001 | $3.57 \times 10^{6}$ | 0.41 | 0.44 |
|  | GO RT | 6.51 | −38.57 | 51.6 | 0.28 | 0.776 | 0.08 | 0.01 | 0.02 |
| 4–5 | PrepRT | 1391.33 | 1132.39 | 1650.27 | 10.96 | <0.001 | $2.48 \times 10^{10}$ | 0.88 | 1.94 |
|  | ACC | 0.06 | −0.01 | 0.13 | 1.66 | 0.107 | 0.64 | 0.32 | 0.29 |
|  | GO RT | −17.41 | −286.67 | 251.84 | −0.13 | 0.896 | 0.19 | 0.02 | 0.02 |
| 6–7 | PrepRT | 1548.91 | 1367.06 | 1730.78 | 17.13 | <0.001 | $5.55 \times 10^{18}$ | 1.21 | 2.47 |
|  | ACC | 0.04 | 0.01 | 0.08 | 2.53 | 0.015 | 2.72 | 0.35 | 0.36 |
|  | GO RT | 39.97 | −32.1 | 112.05 | 1.12 | 0.27 | 0.28 | 0.06 | 0.16 |
| 8–9 | PrepRT | 1263.33 | 1128.5 | 1398.16 | 18.74 | <0.001 | $3.70 \times 10^{23}$ | 0.74 | 2.4 |
|  | ACC | 0.06 | 0.03 | 0.09 | 4.16 | <0.001 | 207.84 | 0.5 | 0.53 |
|  | GO RT | 6.04 | −31.7 | 43.79 | 0.32 | 0.75 | 0.15 | 0.01 | 0.04 |
| 10–11 | PrepRT | 1111.77 | 945.2 | 1278.35 | 13.63 | <0.001 | $3.11 \times 10^{12}$ | 1.0 | 2.45 |
|  | ACC | 0.07 | 0.04 | 0.1 | 4.91 | <0.001 | 764.96 | 0.82 | 0.88 |
|  | GO RT | −6.63 | −45.42 | 32.15 | −0.35 | 0.729 | 0.2 | 0.03 | 0.06 |
| 17–19 | PrepRT | 998.74 | 771.49 | 1225.98 | 8.99 | <0.001 | $1.59 \times 10^{8}$ | 1.26 | 1.64 |
|  | ACC | 0.04 | 0.01 | 0.06 | 2.65 | 0.013 | 3.63 | 0.54 | 0.48 |
|  | GO RT | −7.75 | −41.38 | 25.87 | −0.47 | 0.641 | 0.22 | 0.04 | 0.09 |

diff = difference between long and short preparation trials; lowerCI, upperCI = lower and upper limit of 95% confidence interval; $t$, $p$ = $t$- and $p$-value from paired-samples $t$-test; $BF_{10}$ = Bayes factor, the likelihood of obtaining the current data under the alternative hypothesis, divided by the likelihood of obtaining the current data under the null hypothesis (the default prior Cauchy's width = 0.707 was used); $g_{av}$ = Hedges's average $g$; $d_z$ = Cohen's $d_z$.

was used to study (automatic) effects of instruction implementation. For the purposes of the present study, we further simplified the procedure as we were not interested in the interference effects (in other words, we did not need the NEXT phase). This would allow us to further increase the number of trials on which participants have to execute a novel task for the first time. We tested such a procedure in this experiment.

The simplified procedure consisted of a series of mini-blocks. Each mini-block consisted only of two main phases: an instruction phase and an execution phase (followed by performance feedback). Building further upon the procedure used by Verbruggen *et al.* [10], four novel S-R mappings were presented during the instruction phase, and participants were told to press the space bar when they were ready. We increased the number of S-R mappings from two to four to make the task more challenging, as we were testing adults only. Also, compared to Verbruggen *et al.* [10], there was no longer a minimum preparation duration. The instruction phase thus ended immediately when participants pressed the space bar, regardless of the latency of this response. After the space bar was pressed, the execution phase started. This phase consisted of only one trial, in which one of the four stimuli of the instruction phase was shown. The execution phase consisted of only one trial, as subsequent trials could already be influenced by practice or learning (for the same reason, we focused on the first GO trial in the above reanalysis). By dropping both the NEXT phase and the second GO trial, the duration of each mini-block was substantially reduced, allowing us to increase the total number of mini-blocks (100 mini-blocks instead of 48).

Based on the results of the reanalysis of Verbruggen *et al.* [10] and Cole *et al.* [3], we hypothesized that task performance should be better (higher ACC, and potentially shorter execution RT, exeRT) after long self-paced preparation than after short self-paced preparation.

## 4.1. Methods

### 4.1.1. Participants

All participants were recruited via prolific.co and received £3 for their participation. Participants could sign up for the experiment if they were between 18 and 35 years old, fluent in English, had an approval rate of at least 70% on prolific.co (the approval rate indicates the percentage of prolific submissions of an individual participant that have previously been completed and approved by an experimenter), and if they had not participated in previous studies of this topic hosted by the first author (C.B.R.) on prolific.co.

We tested 100 participants on 24 June 2019. However, one dataset was lost due to server issues. In addition to the participants who completed the experiment and received the monetary compensation, 15 participants started the experiment, but did not complete it and two participants did not complete the experiment during the maximum completion time (i.e. participants had 90 min to finish the experiment). The final sample consisted of 43 females and 56 males ($M_{age} = 25.4$, $s.d._{age} = 4.9$, range 18–36 years).

### 4.1.2. Apparatus and stimuli

The experiment was programmed in jsPsych (v. 6.0.5 [19]). The stimuli were 400 black-and-white images of everyday items, presented against a white background (150 × 150 pixels, taken from [20]). Participants were instructed to respond using the D, F, J and K keys of their keyboard.

### 4.1.3. Procedure

The experiment consisted of 100 mini-blocks and each mini-block consisted of an instruction phase, an execution phase (one trial only) and a feedback screen. The sequence of events is shown in figure 3. At the beginning of each mini-block, the message 'Press space to start a new block' appeared, and participants were asked to press the space bar with one of their thumbs when they felt ready to start a new mini-block. In this instruction phase, four novel S-R mappings were presented. The S-R mappings consisted of four horizontally arranged black-and-white images of everyday objects with the corresponding response keys (the letters D, F, J or K; font size 50 pixels) presented underneath them. (Note that instructions were presented in a similar fashion in the NEXT paradigm used in e.g. Verbruggen *et al.* [10].) Preparation of the S-R mappings was always self-paced, and we told participants that 'You can take as much time as you want to learn the mappings between the images

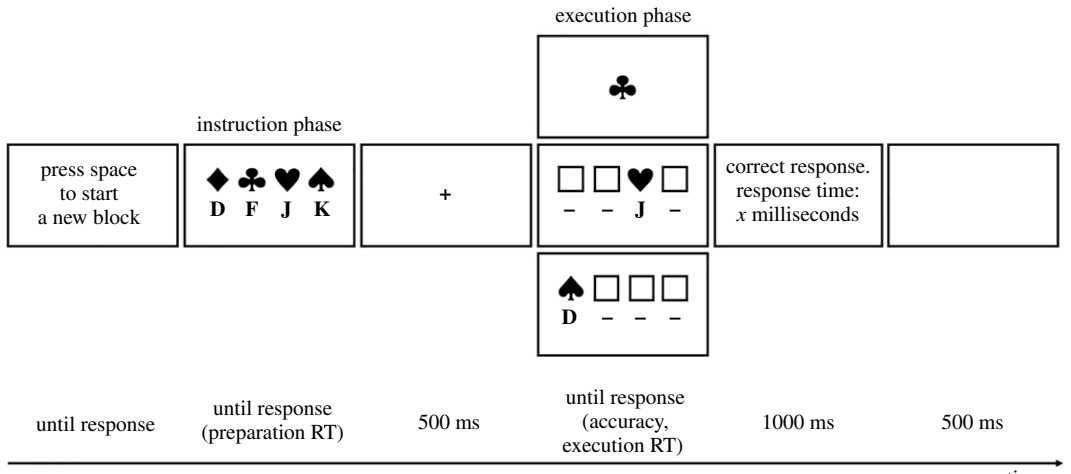

**Figure 3.** Sequence of events in the self-paced preparation paradigm in Experiments 1–4 including an example of each trial type in the execution phase (upper panel: image-alone trials in Experiments 1–4; middle panel: image-and-letter trials in Experiments 2–3; lower panel: image-and-letter trials in Experiment 4). See the text for details.

and the corresponding response keys. When you feel ready, press space with one of your thumbs to start a trial'. The execution phase consisted of only one trial per mini-block, a so-called *image-alone trial*. After a fixation of 500 ms, one image of the instruction phase was presented alone in the centre of the screen and participants had to respond to it according to the prepared S-R mappings. For the execution phase, participants were told that 'Your task is to press the corresponding response key as fast and as accurately as possible'. Across all 100 trials, each of the four response keys was the correct key with equal probability (i.e. 25%). Participants were asked to respond as fast and as accurately as possible by pressing the corresponding keys (D, F, J or K) with their left and right index and middle fingers. After each mini-block, accuracy and the latency of the execution response was displayed for 1000 ms and followed by a blank screen for 500 ms. Participants could take breaks if they wanted to. However, they were specifically instructed to take breaks before the start of a new mini-block (i.e. when the start screen was shown), but not during the self-paced instruction or execution phases. This allowed participants to complete the experiment at their own pace.

After the experiment, participants filled out the UPPS-P Impulsive Behaviour Scale [21]. The questionnaire data were collected in the context of a larger study in which we investigate whether impulsive personality traits are correlated with various types of behaviour. For completeness, we provide the UPPS-P data of the present study in appendix B. Since the individual experiments of the present study do not have enough power for a meaningful analysis of the UPPS-P data [22], we collapsed the UPPS-P data across Experiments 1–4 and reported them together.

### 4.1.4. Dependent variables and analyses

For all trials, mean prepRT, mean ACC and mean exeRT were calculated. Prior to the calculation of mean prepRT and mean ACC, prepRT less than 100 ms and greater than 20 000 ms as well as exeRT less than 100 ms and greater than 3000 ms were excluded (0.99% of trials; we chose these cut-off values after analysing pilot data, which is why they differ from the cut-off values in the reanalysis of Verbruggen *et al.* [10]). Prior to the calculation of mean exeRT, incorrect trials were further excluded (12.59% of trials).

To investigate the link between self-paced preparation and task performance, mean ACC and mean exeRT were again compared between mini-blocks with short and long self-paced preparation, respectively (using the same *pair-split* analysis as outlined above).

## 4.2. Results and discussion

As presented in figure 4, the pair-split analysis revealed that prepRT was 3573 ms for short self-paced preparation and 5123 ms for long self-paced preparation. As shown in table 2, ACC was higher after long self-paced preparation ($M = 0.89$) than after short self-paced preparation ($M = 0.86$, $BF_{10} = 2657$, $p < 0.001$). ExeRT was numerically shorter after long self-paced preparation ($M = 776$ ms) than after

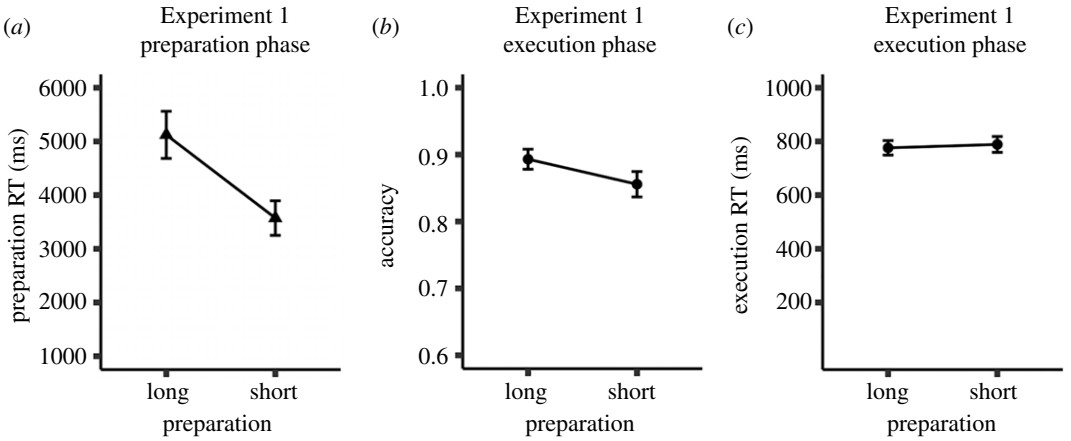

**Figure 4.** Experiment 1, (*a*) mean preparation reaction time (prepRT), (*b*) mean accuracy (ACC) and (*c*) mean execution RT (exeRT) of the pair-split analysis. Error bars stand for 95% within-subject confidence intervals.

short self-paced preparation ($M = 789$ ms), but this difference was not reliable/conclusive ($BF_{10} = 0.59$, $p = 0.06$).

To summarize, task performance was better after long self-paced preparation than after short self-paced preparation. This link was especially pronounced for ACC, as we observed only small numerical differences for exeRT. Overall, the results are in line with the findings of the reanalysis of Verbruggen *et al.* [10] and the findings of Cole *et al.* [3]. Thanks to the courtesy of the authors, we also reanalysed the data from Experiment 3 of Cole *et al.* [3] with our pair-split analysis. We focused exclusively on self-paced preparation and performance in novel tasks when they were encountered for the very first time (thus not practised tasks). The pair-split analysis revealed that mean long self-paced preparation was 3246 ms and mean short self-paced preparation was 2382 ms. Concerning performance, mean ACC was 0.84 after long and 0.83 after short self-paced preparation ($BF_{10} = 0.2$, $p = 0.6$); and mean exeRT was 925 ms after long and 900 ms after short self-paced preparation ($BF_{10} = 0.5$, $p = 0.13$). Thus, our pair-split analysis did not reveal reliable statistical differences in performance between short and long self-paced preparation, which is contrary to the single-trial regression of Cole *et al.* [3]. Considering that Cole *et al.*'s single-trial regression analysis is usually more powerful than a *t*-test on the aggregated data, the absence of an effect could potentially be explained by reduced power in the pair-split analysis. On the other hand, the single-trial regression analysis is more susceptible to global fluctuations in performance, which was controlled for in the pair-split analysis (see appendix A). It should be noted that the individual-difference analysis reported by Cole *et al.* was not confounded by global fluctuations, suggesting that it might indeed be a power issue. Nevertheless, we do not have a conclusive answer for why there is a discrepancy between methods.

Taken together, Experiment 1 indicates that there is indeed a positive link between self-paced preparation time and performance in novel tasks. Now that this positive link is firmly established (at least, using the pair-split method), we can move on to the main question of the study, namely whether this relationship is modulated by a cost–benefit analysis.

# 5. Aim 2: is the duration of self-paced preparation modulated by a cost–benefit analysis?

It has been argued that (self-paced) preparation of novel task instructions (in our case, novel S-R mappings) can be considered as a proactive form of cognitive control [3,23]. Indeed, according to the influential *Dual Mechanisms of Control (DMC) framework* [24,25], proactive control is engaged in anticipation of the requirements of the upcoming task and to prevent conflict between current task goals and distracting information. This can be achieved by actively maintaining task-relevant information in working memory. For example, in our procedure, participants had to process and then maintain the four novel S-R mappings to respond quickly and accurately during the execution phase. Importantly, work within the DMC framework has shown that proactive control comes with both benefits and costs though. Such benefits and costs could have an important influence on for how long

**Table 2.** Statistical results of the pair-split analyses in Experiments 1–4.

| Exp. | measure | long | short | diff | lowerCI | upperCI | $t$ | $p$ | $BF_{10}$ | $g_{av}$ | $d_z$ |
|---|---|---|---|---|---|---|---|---|---|---|---|
| 1 | PrepRT | 5123 (2200) | 3573 (1614) | 1549.66 | 1403.46 | 1695.86 | 21.04 | <0.001 | $5.87 \times 10^{34}$ | 0.81 | 2.11 |
|  | ACC | 0.89 (0.08) | 0.86 (0.1) | 0.04 | 0.02 | 0.05 | 4.8 | <0.001 | $2.66 \times 10^{3}$ | 0.44 | 0.48 |
|  | ExeRT | 776 (135) | 789 (148) | −12.7 | −26.18 | 0.8 | −1.87 | 0.06 | 0.59 | 0.09 | 0.19 |
| 2a | PrepRT | 5108 (2020) | 3673 (1577) | 1435.14 | 1092.47 | 1777.81 | 8.42 | <0.001 | $2.19 \times 10^{9}$ | 0.79 | 1.19 |
|  | ACC | 0.88 (0.09) | 0.86 (0.12) | 0.02 | −0.01 | 0.05 | 1.56 | 0.125 | 0.48 | 0.2 | 0.22 |
|  | ExeRT | 838 (268) | 837 (261) | 0.59 | −29.3 | 30.48 | 0.04 | 0.968 | 0.15 | 0.0 | 0.01 |
| 2b | PrepRT | 4675 (2183) | 3240 (1503) | 1435.61 | 1218.01 | 1653.21 | 13.26 | <0.001 | $7.05 \times 10^{14}$ | 0.77 | 1.88 |
|  | ACC | 0.88 (0.1) | 0.85 (0.11) | 0.02 | −0.1 | 0.05 | 1.94 | 0.058 | 0.86 | 0.24 | 0.27 |
|  | ExeRT | 822 (198) | 876 (222) | −54.74 | −86.96 | −22.52 | −3.41 | 0.001 | 22.61 | 0.26 | 0.48 |
| 3 | PrepRT | 5628 (2735) | 4043 (2247) | 1584.93 | 1400.16 | 1769.7 | 17.01 | <0.001 | $2.75 \times 10^{28}$ | 0.63 | 1.67 |
|  | ACC | 0.86 (0.13) | 0.84 (0.13) | 0.02 | 0.0 | 0.05 | 1.62 | 0.108 | 0.39 | 0.17 | 0.16 |
|  | ExeRT | 834 (237) | 861 (250) | −26.92 | −54.3 | 0.47 | −1.95 | 0.054 | 0.67 | 0.11 | 0.19 |
| 4 | PrepRT | 5878 (2712) | 4084 (2036) | 1793.69 | 1585.14 | 2002.24 | 17.06 | <0.001 | $3.36 \times 10^{28}$ | 0.75 | 1.67 |
|  | ACC | 0.89 (0.1) | 0.85 (0.14) | 0.04 | 0.02 | 0.07 | 3.45 | <0.001 | 26.09 | 0.35 | 0.34 |
|  | ExeRT | 922 (284) | 918 (327) | 4.06 | −28.39 | 36.52 | 0.25 | 0.804 | 0.11 | 0.01 | 0.02 |

long, short = mean preparation reaction time (prepRT, in milliseconds), mean accuracy (ACC, in %), and mean execution RT (exeRT, in milliseconds) in the long and short self-paced preparation trials (standard deviations in parentheses). diff = difference between long and short preparation trials; lowerCI, upperCI = lower and upper limit of 95% confidence interval; $t$, $p$ = $t$- and $p$-value from paired-samples $t$-test; $BF_{10}$ = Bayes factor; $g_{av}$ = Hedges's average $g$; $d_z$ = Cohen's $d_z$

people prepare for upcoming (novel) tasks, *if* such advance (self-paced) preparation is indeed under strategic control.

## 5.1. Benefits of preparation

The benefits of advance (or proactive) task preparation have been observed in several domains (for reviews, e.g. [24]). A full review is beyond the scope of this manuscript, but here we highlight two sets of findings that are particularly relevant for the research on self-paced preparation of novel task instructions.

The study by Cole *et al.* [3] and the findings of the first part of the present study highlight an obvious major benefit of advance preparation, as more elaborate self-paced preparation (as indexed by longer preparation time) seemed to result in more accurate performance (and numerically shorter response execution latencies). Some of the research (using the NEXT paradigm) alluded to before even suggests that (in the extreme), advance preparation might 'remove' the need for further control during the execution phase altogether. This is demonstrated by the so-called *automatic effects of instructions* (AEIs). AEIs refer to the finding that novel tasks that have been instructed may be carried out 'unintentionally' or 'automatically', even though they have never been voluntarily executed nor practised before [26]. As long as the S-R mappings do not change or conflict with each other, such automatic effects (resulting from advance preparation) can be considered as advantageous or beneficial [6,27–30].

Another prominent example regarding the performance benefit of advance preparation comes from research on goal shielding [31,32]. This research demonstrates that advance preparation facilitates goal-directed behaviour, since advance preparation helps focusing on task-relevant information, while reducing interference from irrelevant information. Evidence from dual-task research further corroborates this link. For example, task shielding (i.e. the prioritization of task 1 over task 2 performance) increases with a stronger engagement of proactive control, as indicated by reduced between-task interference [33].

## 5.2. Costs of preparation

Although the performance benefits of advance preparation and proactive control have received a lot of attention, it should not be neglected that advance preparation can also result in performance costs in certain situations [24,34]. This may include e.g. metabolic or time costs. But often, costs and benefits are just opposite sides of the same coin.

For example, the work on AEIs indicates that advance preparation reduces the need for control during the task execution phase. As noted above, this can result in fast and accurate performance, as long as the instructed S-R mappings do not change and the instructed mappings are compatible with the to-be-executed task at hand. However, when the instructed S-R mappings are incompatible or no longer relevant, performance is slower and less accurate, which reflects the cost of advance preparation [6,27–30]. Note that this pattern was also demonstrated in the developmental study that was reanalysed above [10].

Moreover, research on goal-shielding suggests that advance preparation can come with performance costs, especially in ever-changing multi-task environments. Although advance preparation facilitates goal-directed behaviour by shielding the task against irrelevant and unrelated information, performance costs can arise when the 'distracting' information is still somehow related to the task goal, for example, due to a semantic overlap [31,32]. In the extreme, strongly focusing on a particular task could increase the risk that other (even more important or personally relevant) information is not detected or processed [35,36].

Taken together, these examples demonstrate the two sides of advance preparation. Preparation may be beneficial for performance in one situation, but costly in another. People may therefore only prepare for upcoming novel tasks if the cost–benefit analysis is favourable. We tested this general idea in Experiments 2–4, by manipulating the task context and the likelihood that elaborate preparation would be required for successful performance.

## 6. Experiments 2a and 2b

Given that we found evidence for a positive link between self-paced preparation of novel instructions and task performance in the first part of the study, we continued with an investigation of how this link was modulated by benefits and costs of self-paced preparation. To this end, we further modified the

paradigm of Experiment 1 and included the so-called *image-and-letter trials*. All mini-blocks started with an instruction phase, requiring the self-paced preparation of four novel S-R mappings. This was followed by an execution phase, with two possible trial types: image-alone trials or image-and-letter trials. The image-alone trials were the same as the trials of Experiment 1 (i.e. a single image was presented, and participants had to respond by pressing the corresponding key from the instruction phase). On image-and-letter trials, one of the four stimuli of the instruction phase was presented *together* with its associated response key. On all trials, participants had to press the correct response key as fast as possible. Results from the previous experiments indicated that this required advance preparation on image-alone trials. By contrast, on image-and-letter trials, the correct response key was provided on the screen during the execution phase, so that the benefits of advance preparation were reduced. If self-paced preparation is modulated by a cost–benefit analysis, participants should have shorter self-paced preparation time in mini-blocks with image-and-letter trials, compared with mini-blocks with only image-alone trials.

In Experiments 2a and 2b, we varied the probabilities of image-alone and image-and-letter trials between two conditions. Both experiments included a 100%-image-alone condition, in which exclusively image-alone trials were presented (on all trials, participants had to rely on the S-R mappings to respond correctly). The *mixed* condition in Experiment 2a included 50%-image-alone trials and 50%-image-and-letter trials (on which the correct answer was provided on the screen). The *mixed* condition in Experiment 2b included 25%-image-alone trials and 75%-image-and-letter trials. Importantly, in both experiments, participants were not explicitly informed about the probabilities of the trial types. They were simply informed whether only image-alone trials (i.e. 100%-image-alone condition) or both image-alone and image-and-letter trials (i.e. mixed condition) could occur in the upcoming run.

## 6.1. Methods

### 6.1.1. Participants

Participants were tested on prolific.co and received £3 for their participation. They were selected according to the same criteria as in Experiment 1. In both experiments, participants were replaced if they did not finish the experiment and the questionnaire. After excluding prepRT less than 100 ms and greater than 20 000 ms and exeRT less than 100 ms and greater than 3000 ms, participants were replaced if they met one or more of the following criteria: (i) they had less than 50% ACC; (ii) less than 24 image-alone trials left in the 100%-image-alone condition; (iii) less than 12 image-and-letter trials left in the mixed condition (i.e. 50%-image-alone condition) in Experiment 2a; or (iv) less than 18 image-and-letter trials left in the mixed condition (i.e. 25%-image-alone condition) in Experiment 2b.

In Experiment 2a, we tested 50 new participants on 16 October 2019. Two participants did not fulfil the above criteria and were replaced. However, since there were 51 instead of the planned 50 datasets after data collection (probably due to server issues), we only tested one new participant on 9 March 2020. Six participants started the experiment, but did not complete it and another participant did not complete the experiment during the maximum completion time. The final sample consisted of 17 females and 33 males ($M_{age}$ = 25.7, s.d.$_{age}$ = 4.8, range 18–36 years).

In Experiment 2b, we tested 50 new participants on 25 November 2019. Four participants were replaced (two did not finish the experiment and two did not fulfil the criteria; four new participants were tested on 9 March 2020). Seven participants started the experiment, but did not complete it. The final sample consisted of 15 females and 35 males ($M_{age}$ = 26.6, s.d.$_{age}$ = 4.8, range 18–36 years).

### 6.1.2. Apparatus and stimuli

Both experiments were programmed in jsPsych (v. 6.0.5 [19]). In total, 384 images from Experiment 1 were used. The start screen, the instruction phase and the image-alone trials of the execution phase were the same as in Experiment 1. On image-and-letter trials, one image from the instruction phase was selected and presented with the associated response key at the same position as during the instruction phase, with placeholders at the remaining positions (figure 3). For the images, the placeholders were empty squares with a black outline of the same size as the images. For the response keys, the placeholders were dashes.

### 6.1.3. Procedure

The sequence of events in Experiments 2a and 2b is shown in figure 3. The procedure was the same as in Experiment 1 except for the following: there were two trial types in the execution phase. On image-alone trials, after a fixation of 500 ms, an image was presented alone and participants had to respond according to the instructed S-R mappings. On image-and-letter trials, after a fixation of 500 ms, an image was presented together with the correct response key. Image selection was counterbalanced so that in each condition (i.e. image-alone trials in the 100%-image-alone condition, image-alone trials in the mixed conditions, and image-and-letter trials in the mixed conditions), each response key was equally often the correct response key. Participants were asked to respond as fast and as accurately as possible on both image-alone and image-and-letter trials. The rest of the procedure was the same as in Experiment 1.

Both Experiments 2a and 2b contained 96 mini-blocks (thus 96 trials). Experiment 2a contained 48 image-alone trials in the 100%-image-alone condition, and 24 image-alone and 24 image-and-letter trials in the mixed condition (i.e. 50%-image-alone condition). Experiment 2b contained 48 image-alone trials in the 100%-image-alone condition, and 12 image-alone and 36 image-and-letter trials in the mixed condition (i.e. 25%-image-alone condition). In both experiments, the 96 mini-blocks were divided into six runs, with 16 mini-blocks in each run. The conditions alternated between the runs (e.g. 100%-image-alone, mixed, 100%-image-alone, etc.). The condition of the first run was randomized across participants. Prior to each run, participants were informed whether the upcoming run contained only image-alone trials, or both image-alone and image-and-letter trials (the exact probability of image-and-letter trials was not shown). In the mixed conditions, participants did not know in advance whether an upcoming trial would be an image-alone or an image-and-letter trial.

### 6.1.4. Dependent variables and analyses

We used the same data processing procedures as in Experiment 1. Concerning the analysis of prepRT and ACC, this resulted in an exclusion of 3.73% of trials for Experiment 2a, and 1.06% for Experiment 2b. Concerning the analysis of exeRT, this resulted in a further exclusion of 11.26% in Experiment 2a, and 10.87% in Experiment 2b. Performance was again analysed using paired-samples $t$-tests and their Bayesian equivalents. The inferential statistics are reported in table 3. For an overview of the descriptive results, see table 4.

## 6.2. Results

### 6.2.1. Experiment 2a

As shown in figure 5, prepRT in the mixed condition ($M = 4552$ ms) was very similar to prepRT in the 100%-image-alone condition ($M = 4394$ ms), and Bayesian analysis provided 'moderate' evidence for the null hypothesis ($BF_{10} = 0.22$, $p = 0.4$). In other words, participants processed the instructions equally long in both conditions, even though preparation was not required on half of the trials in the mixed condition.

However, we did observe some performance differences on image-alone trials in both conditions. ACC was higher in the 100%-image-alone condition ($M = 0.87$) than in the mixed condition ($M = 0.83$), and there was 'moderate' evidence for this difference ($BF_{10} = 6.33$, $p < 0.001$). Likewise, exeRT was shorter in the 100%-image-alone condition ($M = 838$ ms) than in the mixed condition ($M = 879$ ms) and there was again 'moderate' evidence for this difference ($BF_{10} = 6.02$, $p = 0.01$). Taken together, performance on image-alone trials was overall better in the 100%-image-alone condition compared with the mixed condition, even though prepRT was similar in both conditions. On image-and-letter trials in the mixed condition, ACC was 0.98 (95% CI [0.97, 0.99]) and exeRT was 713 ms (95% CI [655 ms, 771 ms]).

We also performed a pair-split analysis for the 100%-image-alone condition to compare performance between trials with short and long self-paced preparation. These results appear in table 2. PrepRT was 3673 ms for trials with short self-paced preparation and 5108 ms for trials with long self-paced preparation. ACC was 0.88 after long self-paced preparation and 0.86 after short self-paced preparation, but this difference was not significant ($p = 0.12$) and the Bayesian analysis provided 'anecdotal' evidence for the null hypothesis ($BF_{10} = 0.48$). Consistent with Experiment 1, exeRT was similar after short and long self-paced preparation ($M = 837$ ms versus 838 ms, $BF_{10} = 0.15$, $p = 0.97$).

**Table 3.** Comparisons between 100%-image-alone and mixed conditions in Experiments 2a and 2b.

| measure | experiment | diff | lowerCI | upperCI | t | p | BF$_{10}$ | g$_{av}$ | d$_z$ |
|---|---|---|---|---|---|---|---|---|---|
| PrepRT | 2a | −157.94 | −532.02 | 216.14 | −0.85 | 0.4 | 0.22 | 0.08 | 0.12 |
| | 2b | 236.25 | 32.49 | 440.02 | 2.33 | 0.02 | 1.79 | 0.13 | 0.33 |
| ACC | 2a | 0.04 | 0.01 | 0.07 | 2.91 | <0.001 | 6.33 | 0.38 | 0.41 |
| | 2b | 0.1 | 0.06 | 0.14 | 5.0 | <0.001 | 2469.46 | 0.69 | 0.71 |
| ExeRT | 2a | −41.47 | −70.35 | −12.59 | −2.88 | 0.01 | 6.02 | 0.15 | 0.41 |
| | 2b | −65.46 | −108.41 | −22.52 | −3.06 | <0.001 | 9.24 | 0.3 | 0.43 |

diff = difference in mean preparation reaction time (prepRT, in milliseconds), mean accuracy on image-alone trials (ACC, in %), and mean execution RT on image-alone trials (exeRT, in milliseconds) between 100%-image-alone and mixed conditions; lowerCI, upperCI = lower and upper limit of 95% confidence interval; t, p = t- and p-value from paired-samples t-test (in all t-tests, the degrees of freedom were 49); BF$_{10}$ = Bayes factor; g$_{av}$ = Hedges's average g; d$_z$ = Cohen's d$_z$.

**Table 4.** Descriptive results of Experiments 2–4 depending on condition and trial type.

| Exp. | measure | condition | i-a | i-a lowerCI | i-a upperCI | i-a-l | i-a-l lowerCI | i-a-l upperCI |
|------|---------|-----------|-----|-------------|-------------|-------|---------------|---------------|
| 2a | PrepRT | 50% i-a | 4564 (2417) | 3877 | 5251 | 4535 (2519) | 3819 | 5251 |
|    | ACC | 50% i-a | 0.83 (0.13) | 0.79 | 0.86 | 0.98 (0.03) | 0.97 | 0.99 |
|    | ExeRT | 50% i-a | 879 (2809) | 7999 | 959 | 713 (204) | 655 | 771 |
| 2b | PrepRT | 25% i-a | 3704 (1895) | 3166 | 4243 | 3720 (1690) | 3239 | 4200 |
|    | ACC | 25% i-a | 0.76 (0.19) | 0.71 | 0.82 | 0.97 (0.03) | 0.96 | 0.98 |
|    | ExeRT | 25% i-a | 914 (238) | 846 | 981 | 681 (167) | 634 | 729 |
| 3 | PrepRT | 75% i-a | 4792 (2341) | 4336 | 5247 | 4800 (2570) | 4300 | 5300 |
|   | ACC | 75% i-a | 0.84 (0.11) | 0.82 | 0.86 | 0.99 (0.04) | 0.98 | 0.99 |
|   | ExeRT | 75% i-a | 848 (233) | 802 | 893 | 661 (192) | 624 | 699 |
| 3 | PrepRT | 25% i-a | 4256 (2118) | 3844 | 4668 | 4264 (2119) | 3851 | 4676 |
|   | ACC | 25% i-a | 0.78 (0.17) | 0.75 | 0.81 | 0.97 (0.04) | 0.97 | 0.98 |
|   | ExeRT | 25% i-a | 881 (260) | 831 | 932 | 603 (156.6) | 572 | 633 |
| 4 | PrepRT | 75% i-a | 4956 (2233) | 4521.42 | 5390 | 4898 (2306) | 4449 | 5346 |
|   | ACC | 75% i-a | 0.86 (0.1) | 0.84 | 0.89 | 0.93 (0.08) | 0.92 | 0.95 |
|   | ExeRT | 75% i-a | 925 (289) | 869 | 981 | 873 (268) | 821 | 925 |
| 4 | PrepRT | 25% i-a | 4682 (2410) | 4213 | 5150 | 4713 (2309) | 4264 | 5162 |
|   | ACC | 25% i-a | 0.82 (0.15) | 0.79 | 0.85 | 0.95 (0.05) | 0.94 | 0.96 |
|   | ExeRT | 25% i-a | 1014 (366) | 943 | 1085 | 771 (244) | 724 | 819 |

Exp. = Experiment; i-a = image-*alone trials*; i-a-l; = image-*and-letter trials*; lowerCI, upperCI = lower and upper limit of 95% confidence interval.

### 6.2.2. Experiment 2b

In line with our predictions, prepRT was shorter ($M = 3716$ ms) in the mixed condition (where 75% of the trials were image-and-letter trials on which the correct answer was shown) than in the 100%-image-alone condition ($M = 3952$ ms). However, this was only a small effect and the Bayesian analysis provided only 'anecdotal' evidence for the alternative hypothesis ($BF_{10} = 1.79$, $p = 0.02$), as shown in figure 5.

We did observe larger differences in performance though. ACC on image-alone trials was substantially higher in the 100%-image-alone condition ($M = 0.86$) compared with the mixed condition ($M = 0.76$, $BF_{10} = 2469$, $p < 0.001$). Similarly, exeRT on image-alone trials was substantially shorter in the 100%-image-alone condition ($M = 848$ ms) compared with the mixed condition ($M = 914$ ms, $BF_{10} = 9.24$, $p < 0.001$). On image-and-letter trials in the mixed condition, ACC was 0.97 (95% CI [0.96, 0.98]) and exeRT was 681 ms (95% CI [634 ms, 729 ms]).

The pair-split analysis for the 100%-image-alone condition revealed that on average, prepRT on short self-paced preparation trials was 3240 ms and 4675 ms on long self-paced preparation trials (table 2). ACC was 0.88 after long self-paced preparation and 0.85 after short self-paced preparation, but the Bayesian analysis revealed 'anecdotal' evidence for the null hypothesis ($BF_{10} = 0.86$, $p = 0.06$). Moreover, exeRT was shorter after long self-paced preparation ($M = 822$ ms) compared with short self-paced preparation ($M = 877$ ms) and there was 'strong' evidence for this difference ($BF_{10} = 22.61$, $p < 0.001$).

## 6.3. Discussion

In Experiments 2a and 2b, self-paced preparation and performance were contrasted between a condition in which it was clearly beneficial to prepare the novel instructions (i.e. participants had to rely on the instructed S-R mappings on all trials to respond correctly), and a condition in which the benefits of advance preparation were reduced (i.e. the correct response was shown during the execution phase). We therefore anticipated that this would alter the cost–benefit analysis of advance preparation, and specifically, reduce the self-paced preparation time in the mixed conditions compared with the

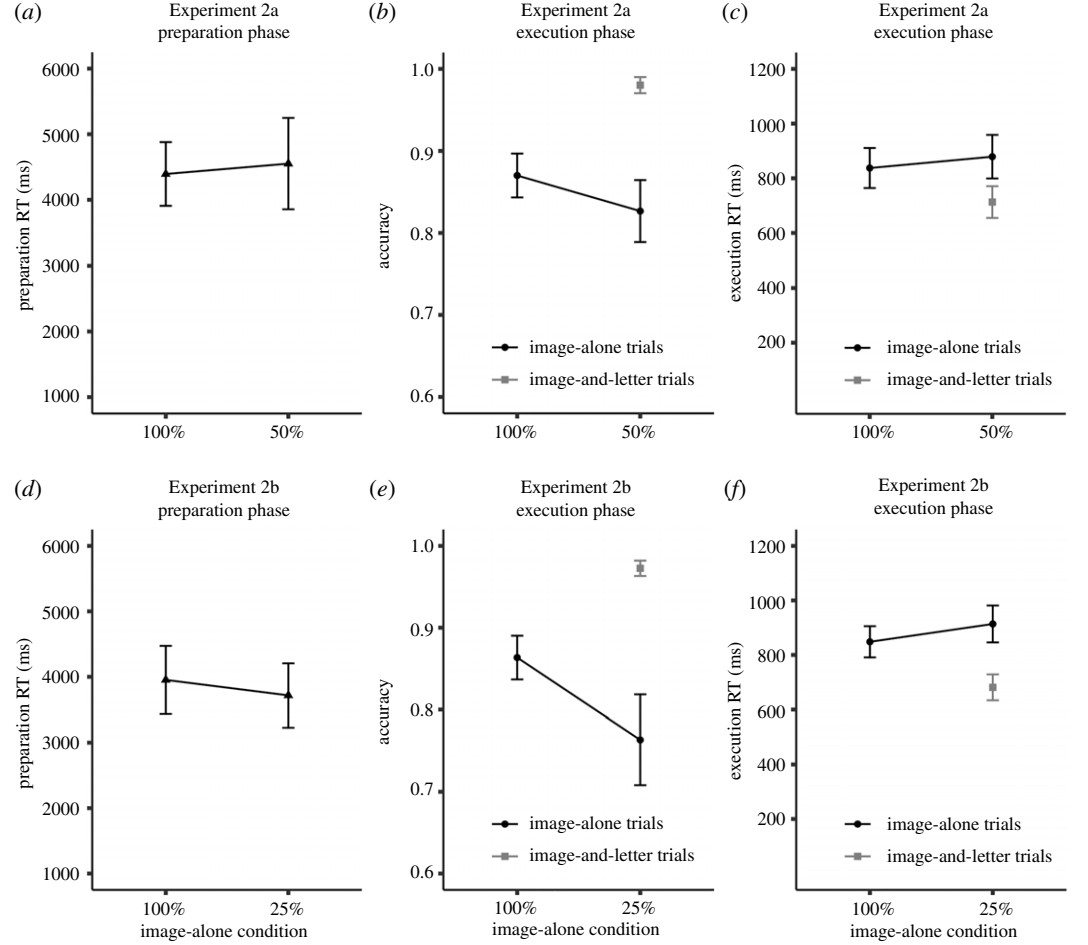

**Figure 5.** Experiments 2a and 2b, (*a,d*) mean preparation reaction time (prepRT), (*b,e*) mean accuracy (ACC), and (*c,f*) mean execution RT (exeRT) depending on condition and trial type. Error bars stand for 95% within-subject confidence intervals.

100%-image-alone condition. But to our surprise, the duration of self-paced preparation did not differ much between both conditions. Rather, participants seemed to prepare the novel instructions more or less equally long irrespective of whether only image-alone trials or both image-alone and image-and-letter trials could be presented (especially in Experiment 2a). A possible explanation for why participants did not adjust the duration of self-paced preparation could be because they did not know the exact probabilities of image-and-letter trials in the mixed conditions. Since participants were not informed about the exact probabilities of image-and-letter trials, they had to learn these probabilities from the experiment. In other words, they had to encounter the mixed conditions first in order to realize that advance preparation was not always required.

Support for this explanation came from an additional exploratory analysis in which we examined how the difference evolved throughout the experiment. In Experiment 2b, we found that the prepRT difference between the 100%-image-alone and the mixed condition (25%-image-alone) *increased* from when participants encountered both conditions for the first time ($M = 91$ ms, $BF_{10} = 0.17$, $p = 0.67$) over the second time ($M = 233$ ms, $BF_{10} = 0.83$, $p = 0.06$) to the third time ($M = 369$ ms, $BF_{10} = 7.3$, $p < 0.001$). In other words, participants shortened the duration of self-paced preparation in the 25%-image-alone condition after experiencing runs with image-and-letter trials. The same exploratory analysis in Experiment 2a revealed that the numeric values of the prepRT difference between the 100%-image-alone and the mixed condition (50%-image-alone) converged from when participants encountered both conditions for the first time ($M = -197$ ms, $BF_{10} = 0.21$, $p = 0.43$) over the second time ($M = -46$ ms, $BF_{10} = 0.16$, $p = 0.76$) to the third time ($M = 9$ ms, $BF_{10} = 0.15$, $p = 0.98$). Different from Experiment 2b though, in Experiment 2a, participants initially prepared longer in the mixed condition at the beginning, but eventually prepared equally long for the mixed and the 100%-image-alone condition. However, this could also be due to reduced preparation in the 100%-image-alone condition over time. We will investigate this issue in more detail in Experiment 3 by informing participants explicitly about the probabilities of image-alone and image-and-letter trials.

Although self-paced preparation was more or less similar in the 100%-image-alone and the mixed conditions, there were some (small/numerical) effects on performance. In both Experiments 2a and 2b, performance on image-alone trials was overall better in the 100%-image-alone condition compared with the mixed conditions. Expectancy effects could account for lower performance on image-alone trials in the mixed conditions though. In both mixed conditions—but especially in the 25%-image-alone condition—image-alone trials were unexpected events that may orient attention away from the to-be-performed task and lead to performance decrements [37]. We will also elaborate more on this in Experiment 3.

The pair-split analyses for the 100%-image-alone condition in both Experiments 2a and 2b revealed that ACC was only numerically higher after long self-paced preparation than after short self-paced preparation. Whereas exeRT was similar after short and long self-paced preparation in Experiment 2a, exeRT was shorter after long self-paced preparation in Experiment 2b. The results were not as clear as in Experiment 1, probably because there were fewer observations per cell in the pair-split analysis (approx. 50 trials per cell in Experiment 1, but only 24 trials in Experiments 2a and 2b).

# 7. Experiment 3

In Experiments 2a and 2b, participants prepared the novel instructions more or less equally long irrespective of whether advance preparation was beneficial for successful performance on most trials. We had predicted a difference in the duration of self-paced preparation. However, the *post hoc* exploratory analyses indicated that the difference emerged after some experience with the mixed condition. Initially, participants may not have adjusted the duration of self-paced preparation in the mixed condition much because they did not immediately realize that advance preparation was not required on a large proportion of trials. We ran Experiment 3 to address this possibility. We made two major changes compared with Experiments 2a and 2b: we explicitly informed participants about the probabilities of the upcoming trial types, and we included both image-alone and image-and-letter trials in all runs with varying probabilities.

First, participants were explicitly informed about the probabilities of image-alone and image-and-letter trials before each run of mini-blocks. Following the exploratory *post hoc* analysis, we predicted that self-paced preparation would be longer for runs in which participants were informed (in advance) that advance preparation was probably required to respond correctly on most trials compared with runs in which advance preparation was probably not required.

Second, in this experiment, we contrasted 25%-image-alone and 75%-image-alone conditions. Both conditions contained image-alone and image-and-letter trials, but the probabilities of both trial types were reversed (i.e. 25%-image-alone trials and 75%-image-and-letter trials or vice versa). In Experiments 2a and 2b, we observed some (numerical) differences in performance on image-alone trials between the two conditions. This could indicate that participants (somewhat) adjusted their preparation strategies after all. However, impaired performance on image-alone trials in the mixed condition could also be due to the fact that such trials were unexpected (especially in Experiment 2b, with 25%-image-alone trials). Indeed, several lines of research indicate that unexpected, infrequent or surprising events can impair performance. For example, Notebaert *et al.* [37] showed that infrequent events are distracting, and can orient attention away from the to-be-performed task and impair performance (see also [38,39]).

Based on this work, we contrasted two accounts in Experiment 3: the 'preparation' account and the 'orienting' account. The 'preparation' account predicts better performance (higher ACC and shorter exeRT) on image-alone trials in the 75%-image-alone compared with the 25%-image-alone condition, since participants should be more likely to prepare and implement the S-R mappings in the former condition. The 'preparation' account predicts similar performance on image-and-letter trials in the 75%-image-alone and the 25%-image-alone conditions, because the correct response is always provided on these trials. By contrast, the 'orienting' account predicts poorer performance for the infrequent trial type (lower ACC and longer exeRT). Thus, for image-alone trials, the 'orienting' account makes the same prediction as the 'preparation' account (higher ACC and shorter exeRT in the 75%-image-alone compared with the 25%-image-alone condition). However, the 'orienting' account predicts the reversed pattern for image-and-letter trials, as performance should be poorer (lower ACC and longer exeRT) on image-and-letter trials in the 75%-image-alone compared with the 25%-image-alone condition.

## 7.1. Methods

Before collecting the data, we preregistered Experiment 3 on OSF on 1 April 2020 (see https://osf.io/r9zjm/). We planned to compare mean prepRT in both conditions (75%-image-alone versus 25%-image-alone) with a paired-samples *t*-test to investigate the effect of the explicit probability information on self-paced preparation. Since we did not provide explicit probability information in Experiments 2a and 2b, we could not use the effect sizes of these experiments in the power analysis of Experiment 3. Instead, in Experiment 3, we followed the recommendation of Brysbaert [22] that Cohen's $d_z = 0.40$ is a good approximation of the average effect size in psychology. Accordingly, the *a priori* power analysis with G*Power [40] indicated that 52 participants were required to achieve 80% power.

After testing 52 new participants as planned, we realized that there was a programming error. At the beginning of the experiment, the condition of the first run (75%-image-alone versus 25%-image-alone) should have been randomized across participants. However, due to a programming error, every participant started with the same condition, the 25%-image-alone condition (i.e. participants 1–52 completed the 25-75-25-75-25-75 sequence). To correct this mistake, we tested 52 new participants, and all of them started with the 75%-image-alone condition (i.e. participants 53–104 completed the 75-25-75-25-75-25 sequence). Experiment 3 thus contained data from 104 participants.

### 7.1.1. Participants

Participants were recruited on prolific.co with the same criteria as in Experiment 1 and received £3 for their participation. They were replaced if they did not finish the experiment and the questionnaires or if they restarted the experiment. After excluding prepRT less than 100 ms and greater than 20 000 ms and exeRT less than 100 ms and greater than 3000 ms, participants were replaced if their ACC was less than 50% on image-alone trials in one of the conditions, or if their ACC was less than 75% on image-and-letter trials in one of the conditions. Participants were further replaced if the number of observations left in one of the cells was fewer than half of the original number of observations (e.g. the 25%-image-alone condition consisted of 12 image-alone trials, so participants would be replaced if they had fewer than six image-alone trials in that condition).

We tested 52 participants on 3 April 2020. Seven participants were replaced on 6 April 2020 (five due to low ACC and two due to missing trials). In addition to these participants, 13 participants started but did not complete the experiment, and one participant did not complete the experiment in time. After detecting the programming error, we tested 52 new participants on 28 April 2020. No participants of this second group had to be replaced. Another eight participants started but did not complete the experiment, and one participant did not complete the experiment in time. The final sample consisted of 46 females, 56 males, one other and one who did not indicate gender ($M_{age} = 25.3$, s.d.$_{age} = 4.9$, range 18–36 years).

### 7.1.2. Apparatus, stimuli and procedure

Experiment 3 contained two conditions. The 75%-image-alone condition consisted of 36 image-alone and 12 image-and-letter trials, and the 25%-image-alone condition consisted of 12 image-alone and 36 image-and-letter trials. For each condition, the 48 mini-blocks were presented in three runs of 16 mini-blocks and the conditions alternated. Half of the participants completed the 25-75-25-75-25-75 sequence, and the other half completed the 75-25-75-25-75-25 sequence. Before each run, participants were explicitly informed about the probability that an image was presented with the correct response key (i.e. the probability of an image-and-letter trial). In each condition, the image-alone and the image-and-letter trials were randomly distributed over all three runs (i.e. all 48 mini-blocks) but not over individual runs.

For exploratory purposes, participants filled out the conscientiousness scale of the Ten-Item Personality Inventory [41], and the conscientiousness scale of the Brief HEXACO Inventory [42]. The results are reported in appendix B.

### 7.1.3. Dependent variables and analyses

We used the same data processing procedures as before. Prior to the analysis of prepRT and ACC, 2.56% of trials were removed and prior to the analysis of exeRT, 9.82% of trials were further excluded.

Performance was again analysed using paired-samples *t*-tests and their Bayesian equivalents. The inferential statistics are reported in table 5. For an overview of the descriptive results, see table 4.

## 7.2. Results

As can be seen in figure 6, prepRT was longer in the 75%-image-alone condition ($M = 4792$ ms) compared with the 25%-image-alone condition ($M = 4261$ ms, $BF_{10} = 116098$, $p < 0.001$). Participants thus used the explicit probability information of image-alone and image-and-letter trials in the upcoming condition and adjusted the duration of self-paced preparation accordingly.

Concerning performance on image-alone trials, ACC was higher in the 75%-image-alone condition ($M = 0.84$) than the 25%-image-alone condition ($M = 0.78$, $BF_{10} = 4934$, $p < 0.001$). In addition, exeRT was numerically shorter in the 75%-image-alone condition ($M = 848$ ms) compared with the 25%-image-alone condition ($M = 881$ ms), but the Bayesian analysis provided only 'anecdotal' support for the difference ($BF_{10} = 1.35$, $p = 0.02$). Taken together, the findings revealed (yet again) a closer link between prepRT and ACC than between prepRT and exeRT.

For performance on image-and-letter trials, the analysis revealed a speed–accuracy trade-off. ACC was higher in the 75%-image-alone condition ($M = 0.99$) compared with the 25%-image-alone condition ($M = 0.97$), and the analysis revealed 'anecdotal' evidence for this effect ($BF_{10} = 2.65$, $p = 0.01$). In other words, participants responded more accurately to the infrequent event, which is opposite to the prediction of the 'orienting' account. At the same time, exeRT was longer in the 75%-image-alone condition ($M = 661$ ms) compared with the 25%-image-alone condition ($M = 603$ ms, $BF_{10} = 1\,750\,778$, $p < 0.001$). Longer and more accurate responses in the 75%-image-alone condition might indicate that participants still relied on the prepared S-R mappings to respond on image-and-letter trials, even though the correct responses were provided.

Consistent with the previous experiments, we performed a pair-split analysis on the image-alone trials from the 75%-image-alone condition as presented in table 2. PrepRT was 4044 ms for short self-paced preparation and 5628 ms for long self-paced preparation. ACC was numerically higher after long self-paced preparation ($M = 0.86$) compared with short self-paced preparation ($M = 0.84$), but the analysis provided 'anecdotal' evidence for the null hypothesis ($BF_{10} = 0.39$, $p = 0.11$). In addition, exeRT was numerically shorter after long self-paced preparation ($M = 834$ ms) than after short self-paced preparation ($M = 861$ ms), but the analysis provided again 'anecdotal' evidence for the null hypothesis ($BF_{10} = 0.67$, $p = 0.05$).

## 7.3. Discussion

In Experiment 3, participants were explicitly informed about the probabilities of image-alone and image-and-letter trials. Self-paced preparation was indeed longer in the 75%-image-alone condition, where it was more beneficial to prepare the novel S-R mappings as compared with the 25%-image-alone condition. This finding thus showed that participants proactively adjusted their self-paced preparation depending on the task requirements [3,24,25].

Moreover, in this experiment we contrasted the 'preparation' and the 'orienting' accounts, which made different predictions regarding performance on image-and-letter trials. Given that ACC was higher on image-and-letter trials in the 75%-image-alone condition compared with the 25%-image-alone condition and thus higher for the infrequent event, the finding provided evidence against the 'orienting' account. Rather, the result pattern suggested that once participants implemented the S-R mappings during the instruction phase, they continued to rely on them on image-and-letter trials, even when this was not strictly necessary. In Experiment 4, we will address this issue by introducing a new image-and-letter trial type, in which it will be costly to apply the prepared S-R mappings.

As in Experiments 2a and 2b, the pair-split analysis indicated again numerically better performance (i.e. higher ACC and shorter exeRT) after long than after short self-paced preparation. Given the low number of trials, however, the statistical power may not have been sufficient to reveal clear statistical evidence for this link.

## 8. Experiment 4

In this final experiment, we manipulated the cost–benefit analysis again and focused on performance costs to complete the picture. In Experiment 1, we investigated exclusively the benefits of self-paced

**Table 5.** Comparisons between 75%-image-alone and 25%-image-alone conditions in Experiment 3.

| measure | trial type | diff | lowerCI | upperCI | $t$ | $p$ | $BF_{10}$ | $g_{av}$ | $d_z$ |
|---|---|---|---|---|---|---|---|---|---|
| PrepRT | all trials | 530.9 | 346.84 | 714.96 | 5.72 | <0.001 | $1.16 \times 10^6$ | 0.24 | 0.56 |
| ACC | image-alone | 0.06 | 0.04 | 0.09 | 4.95 | <0.001 | $4.93 \times 10^3$ | 0.46 | 0.48 |
| | image-and-letter | 0.01 | 0.0 | 0.02 | 2.6 | 0.01 | 2.65 | 0.31 | 0.26 |
| ExeRT | image-alone | −33.63 | −62.58 | −4.68 | −2.3 | 0.02 | 1.35 | 0.14 | 0.23 |
| | image-and-letter | 58.34 | 40.09 | 76.58 | 6.34 | <0.001 | $1.75 \times 10^6$ | 0.33 | 0.62 |

diff = difference in mean preparation reaction time (prepRT, in milliseconds), mean accuracy (ACC, in %), and mean execution RT (exeRT, in milliseconds) between 75%-image-alone and 25%-image-alone conditions; lowerCI, upperCI = lower and upper limit of 95% confidence interval; $t$, $p$ = $t$- and $p$-value from paired-samples $t$-test (in all $t$-tests, the degrees of freedom were 103); $BF_{10}$ = Bayes factor; $g_{av}$ = Hedges's average $g$; $d_z$ = Cohen's $d_z$.

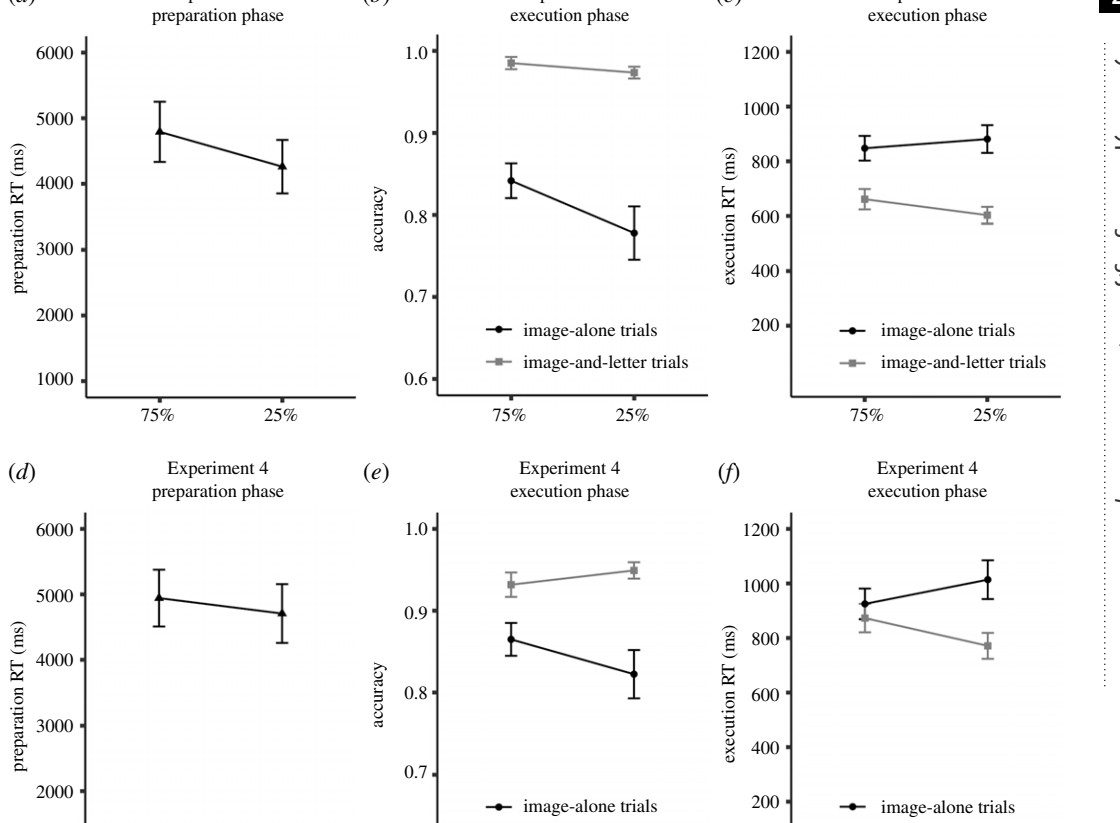

**Figure 6.** Experiments 3 and 4, (*a,d*) mean preparation reaction time (prepRT), (*b,e*) mean accuracy (ACC), and (*c,f*) mean execution RT (exeRT) depending on condition and trial type. Error bars stand for 95% within-subject confidence intervals.

preparation and showed that longer preparation improved performance. In Experiments 2 and 3, we tried to manipulate the benefits of advance preparation by contrasting conditions with different probabilities of image-alone trials. The findings (i.e. especially of Experiment 3) indicated that people adjusted the duration of self-paced preparation depending on the probabilities of image-alone trials. Self-paced preparation was prolonged in the condition with a high probability of image-alone trials and hence higher benefits of advance preparation for accurate performance. However, in Experiments 2 and 3, the performance costs remained more or less the same. Although preparation of the S-R mappings was redundant for performance on image-and-letter trials, it could be argued it was still useful to prepare and apply the S-R mappings, and hence not truly costly. This became especially apparent on image-and-letter trials in Experiment 3 where higher ACC in the 75%-image-alone than the 25%-image-alone condition suggested that participants relied on the prepared S-R mappings to execute the task.

In Experiment 4, we further examined this and altered the cost–benefit analysis. We manipulated the performance costs by creating an image-and-letter task in which preparation of the S-R mappings was truly costly. Specifically, the S-R mappings changed from the instruction phase to the execution phase. In the instruction phase, participants were still asked to prepare four novel S-R mappings. However, in the execution phase of image-and-letter trials, one of the four images was presented, but a new response key was assigned to it and presented on the screen. This new response key was required for correct performance.

In line with Experiment 3, we predicted longer preparation time in the 75%-image-alone compared with the 25%-image-alone condition. We also predicted better performance on image-alone trials in the 75%-image-alone compared with the 25%-image-alone condition, again consistent with Experiment 3. Importantly though, we made different predictions for performance on image-and-letter trials. We predicted that performance on these trials would be worse in the 75%-image-alone

compared with the 25%-image-alone condition as the advance preparation would interfere with the execution of the (incompatible) S-R mappings shown during the execution phase.

## 8.1. Methods

Before collecting the data, we preregistered Experiment 4 on OSF on 26 May 2020 (see https://osf.io/sfxb9/). Since we planned to compare the results of Experiments 3 and 4 in a between-experiment comparison, we also preregistered to test 104 new participants in Experiment 4.

### 8.1.1. Participants

All participants were tested on prolific.co and received £3 for their participation. Participants were selected and replaced based on the same criteria as in Experiment 3. We tested 52 new participants on 2 June 2020 and an extra 52 new participants on 3 June 2020. In total, 29 participants were replaced (27 due to poor performance, 2 due to missing trials). Fourteen participants started the experiment, but did not complete it; two participants did not complete the experiment in time, and one participant was rejected because they completed the experiment within three minutes, which is impossible. The final sample consisted of 43 females and 61 males ($M_{age} = 23.7$, s.d.$_{age} = 4.3$, range 18–35 years).

### 8.1.2. Apparatus, stimuli and procedure

Experiment 4 was identical to Experiment 3, except for a change on image-and-letter trials. On these trials, an image and a letter were shown together, and participants had to press the letter key shown on the screen. Importantly, the image was always associated with a different letter in the instruction phase (figure 3). In other words, the instructed S-R mapping and the S-R mapping shown during the execution phase were always incompatible.

### 8.1.3. Dependent variables and analyses

We used the same data processing procedures as in Experiment 3. Prior to the analysis of prepRT and ACC, 2.85% of trials were removed and prior to the analysis of exeRT, 10.09% of trials were further excluded. Performance was again analysed using paired-samples $t$-tests and their Bayesian equivalents. The inferential statistics are reported in table 6.

For an overview of the descriptive results, see table 4.

## 8.2. Results

As shown in figure 6, prepRT was longer in the 75%-image-alone condition ($M = 4943$ ms) than in the 25%-image-alone condition ($M = 4707$ ms, $BF_{10} = 7.62$, $p < 0.001$). Thus, participants adjusted the duration of self-paced preparation according to the explicit probability information of image-alone trials in the upcoming condition, which is consistent with the findings from Experiment 3.

Performance on image-alone trials was again better in the 75%-image-alone than in the 25%-image-alone condition. ACC was *higher* and exeRT was *shorter* in the 75%-image-alone condition (ACC, $M = 0.86$; exeRT, $M = 925$ ms) compared with the 25%-image-alone condition (ACC, $M = 0.82$; exeRT, $M = 1014$ ms). These differences were reliable (ACC, $BF_{10} = 71.76$, $p < 0.001$; exeRT, $BF_{10} = 290\,996$, $p < 0.001$). Importantly, the opposite pattern was observed on image-and-letter trials. ACC was numerically *lower* and exeRT was *longer* in the 75%-image-alone condition (ACC, $M = 0.93$; exeRT, $M = 873$ ms) compared with the 25%-image-alone condition (ACC, $M = 0.95$; exeRT, $M = 771$ ms). Bayesian analyses provided only anecdotal support for the ACC difference ($BF_{10} = 1.11$, $p = 0.03$), but 'extreme' support for the exeRT difference ($BF_{10} = 9.745648 \times e^{-12}$, $p < 0.001$).

We again performed a pair-split analysis for the image-alone trials of the 75%-image-alone condition (table 2). Short self-paced preparation was 4084 ms and long self-paced preparation was 5878 ms. ACC was higher after long ($M = 0.89$) compared with short self-paced preparation ($M = 0.85$, $BF_{10} = 26.09$, $p < 0.001$). At the same time, exeRT was similar after long ($M = 922$ ms) and short self-paced preparation ($M = 918$ ms), and there was 'moderate' evidence for the null effect ($BF_{10} = 0.11$, $p = 0.8$).

**Table 6.** Comparisons between 75%-image-alone and 25%-image-alone conditions in Experiment 4.

| measure | trial type | diff | lowerCI | upperCI | t | p | BF$_{10}$ | g$_{av}$ | d$_z$ |
|---|---|---|---|---|---|---|---|---|---|
| PrepRT | all trials | 236.22 | 80.9 | 391.54 | 3.02 | <0.001 | 7.62 | 0.1 | 0.3 |
| ACC | image-alone | 0.04 | 0.02 | 0.06 | 3.77 | <0.001 | 71.76 | 0.33 | 0.37 |
| | image-and-letter | −0.02 | −0.03 | 0.0 | −2.21 | 0.03 | 1.11 | 0.27 | 0.22 |
| ExeRT | image-alone | −88.78 | −118.44 | −59.11 | −5.93 | <0.001 | $2.91 \times 10^5$ | 0.27 | 0.58 |
| | image-and-letter | 101.68 | 80.62 | 122.74 | 9.58 | <0.001 | $9.75 \times 10^{12}$ | 0.4 | 0.94 |

diff = difference in mean preparation reaction time (prepRT, in milliseconds), mean accuracy (ACC, in %), and mean execution RT (exeRT, in milliseconds) between 75%-image-alone and 25%-image-alone conditions; lowerCI, upperCI = lower and upper limit of 95% confidence interval; t, p = t- and p-value from paired-samples t-test (in all t-tests, the degrees of freedom were 103); BF$_{10}$ = Bayes factor; g$_{av}$ = Hedges's average g; d$_z$ = Cohen's d$_z$.

**Table 7.** Statistical results of the comparison between Experiments 3 and 4.

| measure and effect | sum-of-squares effect | sum-of-squares residuals | $F(1, 206)$ | $p$ | $\eta_G^2$ | $BF_{incl}$ |
|---|---|---|---|---|---|---|
| *PrepRT* | | | | | | |
| Condition | $1.530 \times 10^{-7}$ | $7.898 \times 10^{-7}$ | 39.91 | <0.001 | 0.01 | $3.64 \times 10^6$ |
| Experiment | $9.241 \times 10^{-6}$ | $2.009 \times 10^{-9}$ | 0.95 | 0.331 | 0.0 | 0.56 |
| Con. × Exp. | $2.258 \times 10^{-6}$ | $7.898 \times 10^{-7}$ | 5.89 | 0.016 | 0.0 | 2.33 |
| *ACC (i-a)* | | | | | | |
| Condition | 0.294 | 1.573 | 38.51 | <0.001 | 0.04 | $2.97 \times 10^6$ |
| Experiment | 0.119 | 6.0 | 4.1 | 0.044 | 0.02 | 1.32 |
| Con. × Exp. | 0.012 | 1.573 | 1.54 | 0.216 | 0.0 | 0.3 |
| *ExeRT (i-a)* | | | | | | |
| Condition | 389 543.602 | $2.340 \times 10^{-6}$ | 34.3 | <0.001 | 0.01 | $3.36 \times 10^5$ |
| Experiment | $1.149 \times 10^{-6}$ | $3.258 \times 10^{-7}$ | 7.27 | 0.008 | 0.03 | 5.36 |
| Con. × Exp. | 79 072.204 | $2.340 \times 10^{-6}$ | 6.96 | 0.009 | 0.0 | 4.11 |
| *ACC (i-a-l)* | | | | | | |
| Condition | $8.853 \times 10^{-4}$ | 0.44 | 0.41 | 0.521 | <0.001 | 0.13 |
| Experiment | 0.157 | 0.734 | 44.01 | <0.001 | 0.12 | $2.41 \times 10^7$ |
| Con. × Exp. | 0.022 | 0.44 | 10.25 | 0.002 | 0.02 | 18.96 |
| *ExeRT (i-a-l)* | | | | | | |
| Condition | 665 740.365 | $1.057 \times 10^{-6}$ | 129.75 | <0.001 | 0.03 | $4.15 \times 10^{19}$ |
| Experiment | $3.759 \times 10^{-6}$ | $1.882 \times 10^{-7}$ | 41.15 | <0.001 | 0.16 | $5.60 \times 10^6$ |
| Con. × Exp. | 48842.708 | $1.057 \times 10^{-6}$ | 9.52 | 0.002 | 0.0 | 11.52 |

Con. = Condition; Exp. = Experiment; i-a = image-*alone* trials; i-a-l = image-*and-letter* trials; *F*, *p* = *F*- and *p*-value of the *F*-statistic; $\eta_G^2$ = effect size generalized eta squared; $BF_{incl}$ = Bayes factor, compares models that contain the effect to equivalent models stripped of the effect.

## 8.3. Comparison between Experiments 3 and 4

In a next step, we directly compared Experiments 3 and 4 in a between-experiment comparison. To further establish if our cost manipulation was indeed successful, we performed a between-experiment analysis for image-and-letter trials to test if the observed difference between experiments (discussed in the previous section) was indeed statistically significant. Then we performed the same between-experiment analysis for prepRT. In Experiment 4, we had predicted (and subsequently observed) longer prepRT in the 75%-image-alone compared with the 25%-image-alone condition. But as we had increased the costs of advance preparation in Experiment 4, we predicted that the between-condition difference should be more pronounced in Experiment 4 compared with Experiment 3. And if the preparation difference would be more pronounced, it also follows that the between-condition difference on image-alone trials should also be more pronounced in Experiment 4 compared with Experiment 3.

### 8.3.1. Dependent variables and analyses

Mean prepRT on all trials, mean ACC and exeRT on image-alone trials, and mean ACC and exeRT on image-and-letter trials were separately analysed with a $2 \times 2$ repeated-measures analysis of variance (ANOVA) and the Bayesian equivalent, with the within-subjects factor condition (25%-image-alone versus 75%-image-alone) and the between-subjects factor experiment (Experiment 3 versus 4). We reported the *p*-value, the Bayes factor $BF_{incl}$ and the effect size generalized eta squared $\eta_G^2$. To follow up a significant interaction, an independent-samples *t*-test and the Bayesian equivalent was conducted. We reported the *p*-value, the Bayes factor $BF_{10}$ and Cohen's *d*. All analyses were performed in JASP (version 0.12.2 [43]). The results are reported in tables 7 and 8.

**Table 8.** Follow-up *t*-tests of the comparison between Experiments 3 and 4.

| measure | trial type | diff | lowerCI | upperCI | *t* | *p* | BF$_{10}$ | $d_z$ |
|---|---|---|---|---|---|---|---|---|
| PrepRT | all | 294.68 | 55.23 | 534.13 | 2.43 | 0.016 | 2.34 | 0.34 |
| ExeRT | image-alone | 55.15 | 13.94 | 96.36 | 2.64 | 0.009 | 3.82 | 0.37 |
| ACC | image-and-letter | 0.03 | 0.01 | 0.05 | 6373.00* | 0.024 | 17.14 | 0.18* |
| ExeRT | image-and-letter | −43.34 | −71.04 | −15.64 | −3.09 | 0.002 | 12.28 | −0.43 |

diff = difference between 75%-image-alone and 25%-image-alone conditions compared between Experiments 3 and 4; lowerCI, upperCI = lower and upper limit of 95% confidence interval; *t*, *p* = *t*- and *p*-value from independent-samples *t*-test (in all *t*-tests, the degrees of freedom were 206); BF$_{10}$ = Bayes factor; $d_z$ = Cohen's $d_z$; * statistic of the Mann–Whitney test; the effect size is given by the rank biserial correlation.

### 8.3.2. Results

We start with reporting the results of the between-experiment comparison focusing on image-and-letter trials, as this analysis can further confirm if we indeed managed to create costs of advance preparation. In Experiment 3, ACC was higher in the 75%-image-alone condition compared with the 25%-image-alone condition ($M = 0.99$ versus $0.97$). This pattern was reversed in Experiment 4, and ACC was lower in the 75%-image-alone compared with the 25%-image-alone condition ($M = 0.93$ versus $0.95$). Indeed, there was 'strong' evidence for a cross-over interaction ($BF_{incl} = 18.96$, $p = 0.002$). The interaction thus illustrated that the S-R mapping change on image-and-letter trials in Experiment 4 induced a performance cost (i.e. reduced ACC), since participants could not rely on the prepared S-R mappings any more. In addition, the induced performance cost also affected exeRT on image-and-letter trials. In Experiment 3, exeRT was 58 ms longer in the 75%-image-alone condition ($M = 661$ ms) compared with the 25%-image-alone condition ($M = 603$ ms). In Experiment 4, exeRT was 102 ms longer in the 75%-image-alone condition ($M = 873$ ms) compared with the 25%-image-alone condition ($M = 771$ ms). The interaction indicated 'strong' evidence for this additional prolongation ($BF_{incl} = 11.52$, $p = 0.002$), which showed that the S-R mapping change in Experiment 4 not only affected ACC on image-and-letter trials, but also the response latencies.

Second, we analysed whether the preparation-time difference was influenced by the induced performance cost. In Experiment 3, prepRT was 531 ms longer in the 75%-image-alone condition ($M = 4792$ ms) compared with the 25%-image-alone condition ($M = 4261$ ms). In Experiment 4, however, prepRT was only 236 ms longer in the 75%-image-alone condition ($M = 4943$ ms) compared with the 25%-image-alone condition ($M = 4707$ ms). The interaction indicated 'anecdotal' evidence for a difference between both experiments ($BF_{incl} = 2.33$, $p = 0.016$). Thus (and to our surprise), the induced performance cost in Experiment 4 did not lead to substantially *shorter* self-paced preparation time in the 25%-image-alone condition. This finding showed that participants adapted to the task requirements in Experiment 4, but not necessarily in an optimal way, since they prepared longer than in Experiment 3 even if it produced performance costs. This is somewhat in line with Meiran *et al.*'s [2] suggestion that participants may have to guess (based on indirect information, such as task difficulty) for how long they should prepare.

Finally, a between-experiment analysis of image-alone trials showed that accuracy was generally higher in Experiment 4 than in Experiment 3 (again consistent with the idea that participants generally prepared longer in Experiment 4). The exeRT results are further in line with this hypothesis, although it should be noted that here, we did find the predicted interaction (i.e. a larger difference between conditions in Experiment 4 compared with Experiment 3).

### 8.4. Discussion

In Experiment 4, we manipulated performance costs to further address the cost–benefit analysis in preparation. We modified the image-and-letter task, so that the S-R mappings changed from the instruction phase to the execution phase. Participants therefore could not rely on the prepared S-R mappings to execute the image-and-letter task in Experiment 4. The analysis of the image-and-letter trials indicated that advance preparation indeed produced costs when the mappings changed. The between-condition comparison of Experiment 4 indicated that this influenced preparation strategies, that is, shorter preparation time when the probability of an image-and-letter trial was high. But to our surprise, this difference was smaller compared with Experiment 3. This suggests that even though participants adjusted their preparation strategies, they may have relied on indirect information such as general task difficulty to do so.

## 9. General discussion

The present study aimed at clarifying the role of self-paced preparation in executing novel tasks. Considering that previous research had produced inconsistent findings [2,3], we first wanted to establish if prolonged self-paced preparation indeed leads to performance benefits when executing novel tasks. We reanalysed data from Verbruggen *et al.* [10] using a pair-split analysis to control for global fluctuations in task performance. We found that both (young and older) children and (late) adolescents performed better in mini-blocks where they had prepared longer. To replicate and extend the findings to adults, we then developed a new paradigm, in which (adult) participants prepared

four novel instructions (four S-R mappings) in a self-paced manner. One stimulus was then presented in the execution phase, and participants had to respond as fast and accurately as possible. We again found that longer self-paced preparation of the S-R mappings led to better performance (especially pronounced for ACC, but statistically less reliable for exeRT). Taken together, performance in novel tasks is improved after longer self-paced preparation of novel instructions (in line with [3]). It is remarkable to see this effect not only in young adults, but already in young children from the age of four onward, despite the fact that their cognitive control system is not fully developed yet.

Building on the positive link between self-paced preparation of novel tasks and performance, we then addressed the question of whether the duration of self-paced preparation is modulated by a cost–benefit analysis. We accomplished this in two steps. In Experiments 2–3, we manipulated performance benefits, while keeping performance costs constant. In Experiment 4, we increased performance costs.

To assess the effect of altered benefits of advance preparation on the self-paced duration of the preparation interval and subsequent performance, we included both image-alone trials and image-and-letter trials in Experiments 2 and 3. On image-and-letter trials, an image was presented together with the response key during the execution phase, rendering the preparation of S-R mappings redundant. We compared conditions with either high or low probabilities of image-alone trials in Experiments 2–3. When the probability of image-alone trials was high (e.g. 100%-image-alone condition in Experiments 2a and 2b, 75%-image-alone condition in Experiment 3), it was beneficial to prepare the S-R mappings. However, when the probability of image-alone trials was relatively low (e.g. 50%-image-alone condition in Experiment 2a, 25%-image-alone condition in Experiments 2b and 3), advance preparation was often no longer required for successful performance, as the correct response was provided on image-and-letter trials. Overall, we found (to varying degrees) that participants somewhat adjusted the duration of self-paced preparation depending on the expected performance benefit. They prepared the S-R mappings longer when the probability of image-alone trials was high. This pattern emerged clearly in Experiment 3, where they were explicitly informed about the exact probabilities, but only subtly in the exploratory *post hoc* analyses in Experiments 2a and 2b, where they might have to learn these probabilities first from the task. Taken together, participants considered the (explicitly provided or acquired) probability information and proactively adjusted the duration of self-paced preparation depending on the task requirements [3,24,25].

In Experiments 2 and 3, we varied performance benefits but kept performance costs the same. Put differently, preparation of the S-R mappings was redundant for performance on image-and-letter trials, but not truly costly. This was revealed in participants' performance on image-and-letter trials in Experiment 3. In this experiment, participants showed better performance (higher ACC) on image-and-letter trials in the 75%-image-alone condition (i.e. when these trials were unexpected/infrequent) compared with the 25%-image-alone condition (i.e. when they were expected/frequent). This suggests that participants may have used the prepared S-R mappings to respond on image-and-letter trials in the 75%-image-alone condition as well, since advance preparation was not truly costly. Note that the ACC increase also allowed us to rule out an 'orienting' account for our between-condition differences, as we observed better performance for unexpected events compared with expected events in the image-and-letter analysis.

To better understand the role of performance costs in the cost–benefit analysis, we modified the image-and-letter task and changed the S-R mappings from the instruction phase to the execution phase in Experiment 4. In this version, preparation of the S-R mappings was costly for performance on image-and-letter trials, because correct performance required a new response key. Our results confirmed this prediction. To some extent, the results of this experiment are in line with studies focusing on AEIs (automatic effects of instructions), in, for example, the NEXT paradigm [7,10,26]. The NEXT effect describes improved performance in the NEXT phase when the prepared S-R mappings for the GO phase are compatible with the required response in the NEXT phase, but impaired performance in the NEXT phase when the S-R mappings are incompatible. Here we also showed that instructions that were prepared but never executed could interfere with performance (although we do not claim that the effects observed here are automatic in nature).

## 9.1. Zooming in on the link between self-paced preparation and performance

In order to learn more about the link between self-paced preparation and performance, Cole *et al.* [3] tested this relationship using a single-trial regression analysis. They showed that longer self-paced preparation was associated with higher ACC. We also found evidence for this link, with the pair-split analysis that controls for global fluctuations in performance. The pair-split analysis additionally

revealed that the link between longer self-paced preparation and shorter exeRT was statistically less reliable.

The strong effect of longer self-paced preparation on ACC together with the weak effect on exeRT seems to reflect a particular characteristic of novel task learning and execution. According to the cognitive model of implementing novel instructions [1], preparing novel S-R mappings should lead to both a declarative and a procedural working memory representation, which could be associated with ACC and exeRT, respectively, and which might influence novel task learning and execution to different degrees (see also [44]). The finding by van't Wout & Jarrold [45] that the use of language is crucial during the early stage of novel task learning further corroborates this idea. Initially, the linguistic or declarative representation of the S-R mappings might be important for performance, but the procedural representation might become more relevant once the task has been practised. This would suggest that in our procedure (in which novel S-R mappings were constantly presented and only executed once) no or only weak procedural representations were formed during the instruction phase. This idea receives also some support from a recent study by Theeuwes and colleagues [46]. The authors investigated the role of physical practice and motor imagery for learning and executing novel S-R mappings. With respect to both ACC and exeRT, participants performed worse in the condition without practice, better after motor imagery, and best after physical practice (see also [47,48]). Thus, without any additional physical or mental practice during the self-paced preparation phase, one might speculate that the effects of prolonged self-paced preparation occur at the declarative level rather than the procedural level and manifest in ACC rather than exeRT. This could explain the strong ACC and the weak exeRT effects in the pair-split analyses in the present study.

The results of the present study strongly support a link between self-paced preparation of novel tasks and performance. While this is in line with the findings of Cole et al. [3], it contradicts the results of Meiran et al. [2], who provided evidence against a link between self-paced preparation of practised tasks and performance. Note, however, that a direct comparison between the present study and the work of Meiran et al. [2] is somewhat difficult, since the present study does not contain any practised tasks. The reason is that we were primarily interested in investigating potential benefits and costs of self-paced preparation for novel tasks specifically (and adding practised tasks would have doubled the duration of the experiment, which was not feasible for practical reasons). Nevertheless, it is important to discuss the role of self-paced preparation for performance in novel and practised tasks. As already remarked by Cole et al. [3], different memory systems are required during self-paced preparation for novel compared with practised tasks. In addition to the above-mentioned dissociation between declarative and procedural memory, it has been argued that while preparation of novel tasks requires the formation of novel S-R mappings in working memory, preparation of practised tasks might primarily consist of the retrieval of S-R mappings from long-term memory [49]. It is hence possible that proactive control processes are more engaged during the formation of novel S-R mappings in working memory and less engaged during the retrieval of S-R mappings from long-term memory. This difference could explain why Meiran et al. [2] or Longman et al. [9] failed to find a link between self-paced preparation and performance in their task-switching study. In a more recent study, Perquin and colleagues [50] also reported that participants did not show performance improvements in practised tasks after self-paced preparation. Longer self-paced preparation for the upcoming trial was in particular associated with poorer performance, hence providing evidence against a positive link between self-paced preparation and performance. However, this finding also raises the possibility that global fluctuations could negatively influence performance in practised tasks. For a better understanding of the link between self-paced preparation and performance in both novel and practised tasks, future studies should take two aspects into account. First, and similar to Cole et al. [3], the studies could present both novel and practised tasks in an intermixed fashion, so that potential effects of global fluctuations should in theory apply to both task types. Second, the studies could also explicitly control for global fluctuations, for example by using the pair-split analysis that we introduced in the present study.

## 9.2. Outlook to future research

The results of the present study—especially of Experiments 3 and 4—clearly showed that participants adapted (to a certain degree) the duration of self-paced preparation according to the task requirements, which had consequences for performance. In general, they prepared longer when preparation was required for performance on a large proportion of trials, but shorter when preparation was required on only a small proportion of trials. This shows block-based adjustments.

But we also found within blocks that longer preparation led to better performance. This prompts the question of *how* participants decide that they are sufficiently prepared for the upcoming task and what information they rely on when making this decision. Current theories on skill learning and metacognition provide only very general answers to this question. For example, according to the triarchic theory of learning [48], three systems are involved in learning to perform novel tasks: a representation system to support associative learning, a cognitive control network to allocate attention during the execution of newly learned behaviour, and a metacognitive system that is highly engaged during the formation of new behaviour and supports the establishment of behavioural routines that are necessary to successfully execute novel tasks. But how exactly the metacognitive system achieves this feat, is not (yet) specified.

## 10. Conclusion

To conclude, we developed a new paradigm to demonstrate that longer self-paced preparation of novel instructions (i.e. in form of S-R mappings) leads to better performance (i.e. higher ACC and faster responses) when they are executed for the very first time. Moreover, the duration of self-paced preparation of novel S-R mappings is strategically adjusted depending on the task requirements. Self-paced preparation is prolonged when it is required on the majority of trials. Longer self-paced preparation in turn leads to performance benefits when the prepared S-R mappings are needed to give the correct response. At the same time, however, longer self-paced preparation leads to performance costs when new S-R mappings instead of the prepared ones are necessary to respond correctly. All in all, the results of the present study provide evidence that self-paced preparation is a form of proactive control that takes into account both benefits and costs of preparation.

Ethics. All new online experiments from the present study were approved by the Ethical Committee of the Faculty of Psychology and Educational Sciences of Ghent University under the license number '2019/31/Christina Reimer'. In all online experiments, participants first received an informed consent, and had to read and agree (by clicking on a button that says 'I agree') before they could start the experiment.
Data accessibility. Experimental scripts and materials, raw data files and data analysis scripts that were used in the present study can be found on OSF (see https://osf.io/vzxfy/).
Authors' contributions. C.B.R. conceptualization, methodology, software, formal analysis, investigation, writing-original draft, writing-review and editing; Z.C. methodology, software, formal analysis, writing-review and editing; F.V. conceptualization, methodology, writing-review and editing, funding acquisition.
Competing interests. We declare we have no competing interests.
Funding. This work was supported by an ERC Consolidator grant awarded to Frederick Verbruggen (European Union's Horizon 2020 research and innovation program, grant agreement no. 769595).

## Appendix A

Performance generally improves over practice (i.e. participants become faster and more accurate), even when new S-R mappings are introduced in each block. This was also observed in the original study of Verbruggen *et al.* (see fig. 3 of the original paper [10]). However, such a global improvement could mask preparation-specific effects at a 'local' level (i.e. the effects we are interested in). This is demonstrated in figure 7, depicting some hypothetical data. When visually inspecting global trends (left and middle part of the figure), there appears to be a strong negative correlation: when preparation time decreases, accuracy increases. However, when zooming in and looking at local trends (as illustrated in the right part of the figure), there appears to be a positive correlation, that is, higher accuracy for longer preparation times. Thus, global trends could mask or even confound local trends if not properly controlled for (for similar arguments, e.g. [51,52]). Here we used the pair-split analysis to control for such trends or fluctuations.

## Appendix B

After Experiments 1–4, participants filled out the UPPS-P [21]. The questionnaire data were collected in the context of a larger study in which we investigate whether impulsive personality traits are correlated with various types of behaviour. Here, we reported the correlations between the subscales of the UPPS-P and mean prepRT for all 407 participants of the present study. Note that we used only prepRT of the 100%-image-alone conditions in Experiments 1, 2a and 2b, and prepRT of the 75%-image-alone

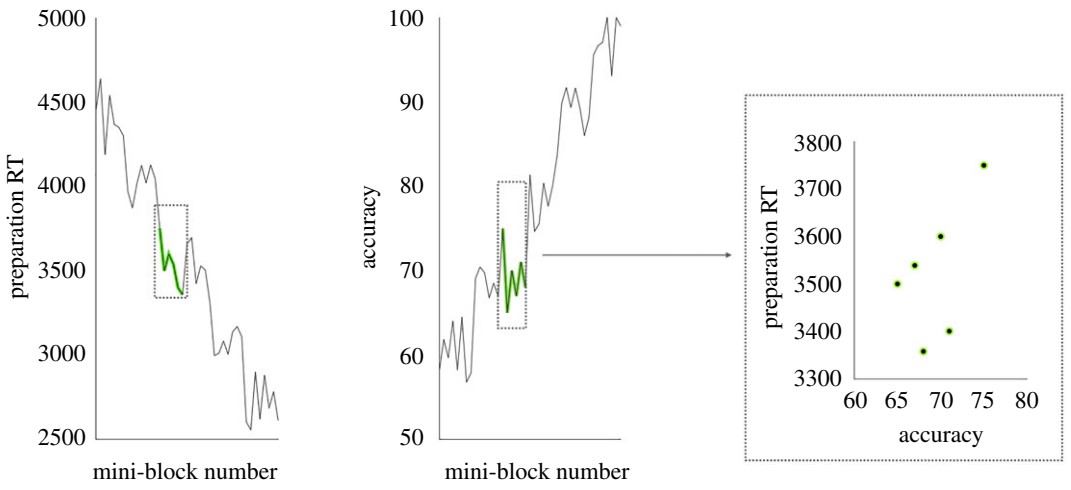

**Figure 7.** Hypothetical global and local trends of preparation time and accuracy over the course of an experiment.

**Table 9.** Correlations between the subscales of the UPPS-P and prepRT.

| UPPS-P subscales | PrepRT | lowerCI | upperCI | *t* | *p* |
|---|---|---|---|---|---|
| negative urgency | −0.04 | −0.14 | 0.06 | −0.81 | 0.417 |
| lack of premeditation | −0.05 | −0.14 | 0.05 | −0.91 | 0.363 |
| lack of perseverance | −0.07 | −0.17 | 0.03 | −1.39 | 0.165 |
| sensation seeking | 0.0 | −0.10 | 0.10 | 0.01 | 0.990 |
| positive urgency | −0.07 | −0.16 | 0.03 | −1.31 | 0.192 |

lowerCI, upperCI = lower and upper limit of 95% confidence interval; *t*, *p* = *t*- and *p*-value from *t*-test (in all *t*-tests, the degrees of freedom were 405).

**Table 10.** Correlations between the conscientiousness subscales of the TIPI and the BHI and prepRT.

| conscientiousness subscales | PrepRT | lowerCI | upperCI | *t* | *p* |
|---|---|---|---|---|---|
| TIPI | −0.03 | −0.16 | 0.11 | −0.36 | 0.716 |
| BHI | −0.03 | −0.16 | 0.11 | −0.39 | 0.694 |

lowerCI, upperCI = lower and upper limit of 95% confidence interval; *t*, *p* = *t*- and *p*-value from *t*-test (in all *t*-tests, the degrees of freedom were 206).

conditions in Experiments 3 and 4, because these conditions relied strongly on self-paced preparation of the novel instructions. As can be seen in table 9, none of the correlations was significant.

Furthermore, in Experiments 3 and 4, participants filled out the conscientiousness subscales of the TIPI [41] and the BHI [42]. As for the UPPS-P, we used only mean prepRT of the 75%-image-alone conditions in both experiments for all 208 participants. As reported in table 10, the correlations between prepRT and the conscientiousness scores were not significant.

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
