## [Peer Review File · Royal Society Open Science]

Review History

RSOS-210762.R0 (Original submission)

Review form: Reviewer 1

Is the manuscript scientifically sound in its present form?

Yes

Are the interpretations and conclusions justified by the results?

Yes

Is the language acceptable?

Yes

Do you have any ethical concerns with this paper?

No

Have you any concerns about statistical analyses in this paper?

No

Recommendation?

Major revision is needed (please make suggestions in comments)

Comments to the Author(s)

The manuscript includes a thorough and comprehensive investigation of self-paced preparation and its relation to task performance, as well as its modulation by costs and benefits of task preparation. This issue was raised in the literature in the recent years, but it was not investigated thoroughly until now, and therefore it suggests an interesting and timely question.

Although the manuscript does offer significant novelty, I found some questions were raised during reading, and some issues require further clarifications.

1. First, In Verbruggen et al. (2018) reanalysis, the authors address the 3s time limit on instructions screens, but report a shorter than 3 seconds preparation for short self-paced miniblocks. This issue should be further clarified in the manuscript.
2. Also in the reanalysis, the authors mentions that the link between self-paced preparation and task performance was evident in all age groups, but it should be more clearly emphasized that it was only robust from age 8, to my understanding.
3. To me, the benefit of the pair-split procedure over the single-trial regression was not clear enough, and the authors are encouraged to elaborate on this issue.
4. The issue of speed-accuracy tradeoff that appears on some of the experiments does not receive proper consideration. It could be considered as raising a problem in interpreting the results, and thus the authors are encouraged to at least speculate on why this pattern showed up in the results.
5. Did participants receive any instructions regarding self-paced instructions keypress? For example, were they at some level encouraged to quickly respond when they feel ready to begin the task execution?
6. Exp 2A - how do the authors explain execution differences without prepRT differences? (Table 3)
7. The authors do not differentiate prepRT for picture-only and picture-and-letter trials. While of course participants could not know in advance which type of trial will follow, a glimpse into this difference could shed light on the latter execution differences. One solution could be to add a descriptive plot (without statistical inference), which will show the relationship between prepRT and ACC/exeRT over trials in each experiment. I believe that such a plot, if informative, could improve the readers' understanding of the relationship between prepRT and task execution.
8. Finally, the abstract is very specific in referring to the different conditions by name, which is, at least to me, very confusing before the conditions were properly introduced in the main text. Thus, I would recommend writing in more general terms that would be more easily understood by the (then) naïve reader.

Review form: Reviewer 2

Is the manuscript scientifically sound in its present form?

Yes

Are the interpretations and conclusions justified by the results?

Yes

Is the language acceptable?

Yes

Do you have any ethical concerns with this paper?

No

Have you any concerns about statistical analyses in this paper?

No

Recommendation?

Accept with minor revision (please list in comments)

Comments to the Author(s)

The present study involved four experiments (as well as reanalysis of two datasets) to improve understanding of the role of self-paced preparation in instructed task learning performance. This question has implications for understanding how (or whether) proactive control is involved in novel task learning. Overall I found that study to be well-executed, and I had only relatively minor concerns, listed below.

Minor concerns:

- In the first data reanalysis, it is unclear from the text how long participants prepared in the NEXT paradigm, since the NEXT phase occurred between the instructions and GO trials. Is the prepRT based on the time viewing the instruction screen, the time during the NEXT phase, or both? Or perhaps NEXT mini-blocks were not used for analysis (assuming there are GO-only mini-blocks)?
- It is noted that the results of Experiment 1 did not replicate in a reanalysis of the Cole et al. 2018 study. It would be useful to the reader to note that the Cole et al. 2018 study also included an individual difference analysis revealing a positive relationship between preparation time and accuracy that was not susceptible to the within-subject "global fluctuations in performance" confound. (Or if the authors think that individual differences are susceptible to that confound it would be good to include a discussion of why.)
- It was noted that the Cole et al. 2018 study included a well-practiced task control condition, which did not show the positive prepRT-ACC relationship. It should be discussed whether this acts as a proper control for the within-subject "global fluctuations in performance" confound mentioned in the present study. It appears that it might be a proper control, given that the well-practiced tasks were intermixed with the novel tasks (such that global performance improvements over the session should in theory apply to both). Such a discussion would be especially important given that it could potentially resolve the discrepancy emphasized throughout the paper: that preparation time has tended to have a negative association with performance of well-practiced tasks (in contrast to a positive relationship for novel tasks). It is unclear why such a control condition was not included in any of the 4 experiments here (especially given the emphasis on the discrepancy in the literature), unless there were some concerns by the authors that such a control condition was not appropriate for some reason.

Decision letter (RSOS-210762.R0)

Dear Dr Reimer

On behalf of the Editors, we are pleased to inform you that your Manuscript RSOS-210762 "Benefits and costs of self-paced preparation of novel task instructions" has been accepted for publication in Royal Society Open Science subject to minor revision in accordance with the referees' reports. Please find the referees' comments along with any feedback from the Editors below my signature.

Please submit your revised manuscript and required files (see below) no later than 7 days from today's (ie 06-Aug-2021) date. Note: the ScholarOne system will 'lock' if submission of the revision is attempted 7 or more days after the deadline. If you do not think you will be able to meet this deadline please contact the editorial office immediately.

on behalf of Professor Zoltan Dienes (Associate Editor) and Essi Viding (Subject Editor)
openscience@royalsociety.org

Associate Editor Comments to Author (Professor Zoltan Dienes):

I now have two reviews for your paper, which are largely positive; for your resubmission I will assess the extent to which you have dealt with their points.

Reviewer comments to Author:

Reviewer: 1

Comments to the Author(s)

The manuscript includes a thorough and comprehensive investigation of self-paced preparation and its relation to task performance, as well as its modulation by costs and benefits of task preparation. This issue was raised in the literature in the recent years, but it was not investigated thoroughly until now, and therefore it suggests an interesting and timely question.

Although the manuscript does offer significant novelty, I found some questions were raised during reading, and some issues require further clarifications.

1. First, In Verbruggen et al. (2018) reanalysis, the authors address the 3s time limit on instructions screens, but report a shorter than 3 seconds preparation for short self-paced miniblocks. This issue should be further clarified in the manuscript.
2. Also in the reanalysis, the authors mentions that the link between self-paced preparation and task performance was evident in all age groups, but it should be more clearly emphasized that it was only robust from age 8, to my understanding.
3. To me, the benefit of the pair-split procedure over the single-trial regression was not clear enough, and the authors are encouraged to elaborate on this issue.
4. The issue of speed-accuracy tradeoff that appears on some of the experiments does not receive proper consideration. It could be considered as raising a problem in interpreting the results, and thus the authors are encouraged to at least speculate on why this pattern showed up in the results.
5. Did participants receive any instructions regarding self-paced instructions keypress? For example, were they at some level encouraged to quickly respond when they feel ready to begin the task execution?
6. Exp 2A - how do the authors explain execution differences without prepRT differences? (Table 3)
7. The authors do not differentiate prepRT for picture-only and picture-and-letter trials. While of course participants could not know in advance which type of trial will follow, a glimpse into this difference could shed light on the latter execution differences. One solution could be to add a descriptive plot (without statistical inference), which will show the relationship between prepRT and ACC/exeRT over trials in each experiment. I believe that such a plot, if informative, could improve the readers' understanding of the relationship between prepRT and task execution.
8. Finally, the abstract is very specific in referring to the different conditions by name, which is, at least to me, very confusing before the conditions were properly introduced in the main text. Thus, I would recommend writing in more general terms that would be more easily understood by the (then) naïve reader.

Reviewer: 2

Comments to the Author(s)

The present study involved four experiments (as well as reanalysis of two datasets) to improve understanding of the role of self-paced preparation in instructed task learning performance. This question has implications for understanding how (or whether) proactive control is involved in novel task learning. Overall I found that study to be well-executed, and I had only relatively minor concerns, listed below.

Minor concerns:

- In the first data reanalysis, it is unclear from the text how long participants prepared in the NEXT paradigm, since the NEXT phase occurred between the instructions and GO trials. Is the prepRT based on the time viewing the instruction screen, the time during the NEXT phase, or both? Or perhaps NEXT mini-blocks were not used for analysis (assuming there are GO-only mini-blocks)?
- It is noted that the results of Experiment 1 did not replicate in a reanalysis of the Cole et al. 2018 study. It would be useful to the reader to note that the Cole et al. 2018 study also included an individual difference analysis revealing a positive relationship between preparation time and accuracy that was not susceptible to the within-subject "global fluctuations in performance"

confound. (Or if the authors think that individual differences are susceptible to that confound it would be good to include a discussion of why.)

- It was noted that the Cole et al. 2018 study included a well-practiced task control condition, which did not show the positive prepRT-ACC relationship. It should be discussed whether this acts as a proper control for the within-subject "global fluctuations in performance" confound mentioned in the present study. It appears that it might be a proper control, given that the well-practiced tasks were intermixed with the novel tasks (such that global performance improvements over the session should in theory apply to both). Such a discussion would be especially important given that it could potentially resolve the discrepancy emphasized throughout the paper: that preparation time has tended to have a negative association with performance of well-practiced tasks (in contrast to a positive relationship for novel tasks). It is unclear why such a control condition was not included in any of the 4 experiments here (especially given the emphasis on the discrepancy in the literature), unless there were some concerns by the authors that such a control condition was not appropriate for some reason.

===PREPARING YOUR MANUSCRIPT===

===PREPARING YOUR REVISION IN SCHOLARONE===

Author's Response to Decision Letter for (RSOS-210762.R0)

See Appendix A.

Decision letter (RSOS-210762.R1)

Dear Dr Reimer,

I am pleased to inform you that your manuscript entitled "Benefits and costs of self-paced preparation of novel task instructions" is now accepted for publication in Royal Society Open Science.

on behalf of Professor Zoltan Dienes (Associate Editor) and Essi Viding (Subject Editor)
openscience@royalsociety.org

Follow Royal Society Publishing on Twitter: @RSocPublishing
Follow Royal Society Publishing on Facebook:
<https://www.facebook.com/RoyalSocietyPublishing.FanPage/>

Read Royal Society Publishing's blog:
<https://royalsociety.org/blog/blogsearchpage/?category=Publishing>

Appendix A

Frederick Verbruggen
Professor of Experimental Psychology
E: frederick.verbruggen@ugent.be
T: +32 9 264 64 41
H. Dunantlaan 2
B-9000 Ghent
Belgium
cobe.ugent.be

To:
Professor Zoltan Dienes, Associate Editor, and Essi Viding, Subject Editor
Royal Society Open Science
RSOS-210762

Dear Professor Zoltan Dienes, Dear Essi Viding,

We would like to submit a revised version of our manuscript entitled 'Benefits and costs of self-paced preparation of novel task instructions' to Royal Society Open Science.

We thank you and the two reviewers for the positive overall evaluation as well as for the valuable comments and suggestions on how to further improve the manuscript. We addressed every point in the revision and we believe that these changes improved both the strength and clarity of the manuscript.

Below you will find our point-by-point responses to the reviewers' comments. We hope that the changes we made will meet your and the reviewers' expectations and we look forward to hearing from you.

Best regards,

Christina Reimer (in the name of the co-authors)

Reviewer: 1

Comments to the Author(s)

The manuscript includes a thorough and comprehensive investigation of self-paced preparation and its relation to task performance, as well as its modulation by costs and benefits of task preparation. This issue was raised in the literature in the recent years, but it was not investigated thoroughly until now, and therefore it suggests an interesting and timely question. Although the manuscript does offer significant novelty, I found some questions were raised during reading, and some issues require further clarifications.

Response: We would like to thank the reviewer for the positive evaluation of our manuscript. We also appreciate the helpful comments, which helped us to further improve our manuscript.

1. First, in Verbruggen et al. (2018) reanalysis, the authors address the 3s time limit on instructions screens, but report a shorter than 3 seconds preparation for short self-paced mini-blocks. This issue should be further clarified in the manuscript.

Response: We apologize for the confusion. In the NEXT paradigm, the novel instructions stayed on screen for at least 3 seconds. However, during the 3 seconds interval, participants could at any time indicate via key press that they finished preparing the instructions already, which explains that prepRT was shorter than 3 seconds on some trials. In that case, participants could only start with the NEXT phase once the 3 seconds had fully elapsed to ensure that each participant was presented with the instructions for at least 3 seconds. On the other hand, if participants needed more time than 3 seconds to prepare the instructions, the instructions simply stayed on screen until participants finished preparation, and they could immediately proceed with the NEXT phase. We clarified this issue in the manuscript on p. 86: "In the self-paced instruction phase, the instructions remained on screen until participants had pressed a key *and* at least 3 seconds had elapsed. If they pressed the start key before 3 seconds had elapsed, the response was registered (explaining why prepRT was sometimes shorter than 3 seconds), but they had to wait a bit before the NEXT phase started. This was done to ensure that individuals of all age groups would process the instructions to some extent. Note that if participants pressed the start key after 3 seconds, the NEXT phase would start immediately."(p. 86)

2. Also in the reanalysis, the authors mention that the link between self-paced preparation and task performance was evident in all age groups, but it should be more clearly emphasized that it was only robust from age 8, to my understanding.

Response: We agree with the reviewer's comment, so we changed the interpretation of the results accordingly on p. 88: "Across participants of all age groups, ACC was (numerically) higher after long self-paced preparation than after short self-paced preparation. It should be noted though that the statistical analysis revealed that this effect was only robust from age group 8-9 onward."(p. 88)

3. To me, the benefit of the pair-split procedure over the single-trial regression was not clear enough, and the authors are encouraged to elaborate on this issue.

Response: We added a new figure with hypothetical data in Appendix 1 to illustrate the benefit of the pair-split analysis. Performance generally improves over practice (i.e., participants become faster and more accurate), even when new S-R mappings are introduced in each block. Note that this was also observed in the original study of Verbruggen et al. (2018), as shown in Figure 3 of that paper. However, such a global improvement could mask preparation-specific effects at a 'local' level (i.e. the effects we are interested in). This is demonstrated in the new figure: When visually inspecting global trends (left and middle part of the figure), there appears to be a strong negative correlation: when preparation time decreases, accuracy increases. However, when zooming in and looking at local trends (as illustrated in the right part of the figure), there appears to be a positive correlation, that is, higher accuracy for longer preparation times. Thus, we used the pair-split analysis to control for global trends/fluctuations that might mask or even confound local effects of preparation (pp. 86-87, Reanalysis of Verbruggen et al. (2018), and p. 114, Appendix 1).

4. The issue of speed-accuracy trade-off that appears on some of the experiments does not receive proper consideration. It could be considered as raising a problem in interpreting the results, and thus the authors are encouraged to at least speculate on why this pattern

showed up in the results.

Response: Unfortunately, it is not clear to us to which experiment(s) the reviewer is referring to. Perhaps the reviewer is referring to performance on image-and-letter trials in Experiment 3 (p. 102)? Indeed, in this experiment, it seemed that participants responded slower and more accurate on image-and-letter trials in the 75%-image-alone condition compared with the 25%-image-alone condition. However, we addressed this speed/accuracy issue in the discussion of Experiment 3 (p. 102). Importantly, our proposed explanation for the trade-off was even tested explicitly in Experiment 4, where we changed the S-R mappings between the instruction and the execution phase. No other speed/accuracy trade-offs were observed in the experiments (apart from a 3 ms difference in the pair-split analyses in Experiment 4; but note again that in this experiment, we explicitly manipulated the costs of advance preparation). We hope that our response addresses the reviewer's concern sufficiently.

5. Did participants receive any instructions regarding self-paced instructions keypress? For example, were they at some level encouraged to quickly respond when they feel ready to begin the task execution?

Response: In the beginning of all experiments (thus before starting the proper experimental trials), participants were told the following about the (S-R mapping) instruction phase: "You can take as much time as you want to learn the mappings between the images and the corresponding response keys. When you feel ready, press space with one of your thumbs to start a trial." The instruction emphasized that preparation of the novel S-R mappings was entirely voluntary so that participants could decide for themselves how quickly they wanted to do the task. Afterwards, for the execution phase, participants were told that "Your task is to press the corresponding response key as fast and as accurately as possible." Here, we encouraged participants to execute the novel task as fast and as accurately as they could. We added the precise instructions for both the instruction and the execution phase in the methods part of Experiment 1, please see p. 90.

6. Exp 2A - how do the authors explain execution differences without prepRT differences? (Table 3)

Response: We would like to refer the reviewer to pp. 98-99, where we discussed a potential explanation for this observation. Specifically, we wrote the following on p. 99: "Although self-paced preparation was more or less similar in the 100%-image-alone and the mixed conditions, there were some (small/numerical) effects on performance. In both Experiments 2a and 2b, performance on image-alone trials was overall better in the 100%-image-alone condition compared to the mixed conditions. Expectancy effects could account for lower performance on image-alone trials in the mixed conditions though. In both mixed conditions - but especially in the 25%-image-alone condition - image-alone trials were unexpected events that may orient attention away from the to-be-performed task and lead to performance decrements (Notebaert et al., 2009). We will also elaborate more on this in Experiment 3." As noted, we tested this idea in Experiment 3.

7. The authors do not differentiate prepRT for picture-only and picture-and-letter trials. While of course participants could not know in advance which type of trial will follow, a glimpse into this difference could shed light on the latter execution differences. One solution could be to add a descriptive plot (without statistical inference), which will show the relationship between prepRT and ACC/exeRT over trials in each experiment. I believe that such a plot, if informative, could improve the readers' understanding of the relationship between prepRT

and task execution.

Response: We appreciate the reviewer's suggestion and added a table that shows the descriptive results of Experiments 2 - 4 depending on condition and trial type. As shown in Table 4, prepRT are very similar for image-alone and image-and-letter within the same condition (p. 98).

8. Finally, the abstract is very specific in referring to the different conditions by name, which is, at least to me, very confusing before the conditions were properly introduced in the main text. Thus, I would recommend writing in more general terms that would be more easily understood by the (then) naïve reader.

Response: We revised the abstract accordingly and replaced the names of the trial types with more general terms (p. 83).

Reviewer: 2

Comments to the Author(s)

The present study involved four experiments (as well as reanalysis of two datasets) to improve understanding of the role of self-paced preparation in instructed task learning performance. This question has implications for understanding how (or whether) proactive control is involved in novel task learning. Overall I found that study to be well-executed, and I had only relatively minor concerns, listed below.

Response: We are very happy about the positive evaluation of our manuscript. We are also grateful for the helpful comments, which helped us to further improve our manuscript.

Minor concerns: 1. In the first data reanalysis, it is unclear from the text how long participants prepared in the NEXT paradigm, since the NEXT phase occurred between the instructions and GO trials. Is the prepRT based on the time viewing the instruction screen, the time during the NEXT phase, or both? Or perhaps NEXT mini-blocks were not used for analysis (assuming there are GO-only mini-blocks)?

Response: In the NEXT paradigm, prepRT was measured from onset of the instruction screen until participants indicated via key press that they finished preparing the instructions. After the instruction phase, participants had to advance through the NEXT phase, and they eventually applied the prepared instructions in the GO phase. Thus, participants had to maintain the instructions in memory during the NEXT phase, so that they could execute them in the GO phase. We clarified this issue in the manuscript on p. 86: "In the self-paced instruction phase, the instructions remained on screen until participants had pressed a key *and* at least 3 seconds had elapsed. If they pressed the start key before 3 seconds had elapsed, the response was registered (explaining why prepRT was sometimes shorter than 3 seconds), but they had to wait a bit before the NEXT phase started. This was done to ensure that individuals of all age groups would process the instructions to some extent. Note that if participants pressed the start key after 3 seconds, the NEXT phase would start immediately."

2. It is noted that the results of Experiment 1 did not replicate in a reanalysis of the Cole et al. 2018 study. It would be useful to the reader to note that the Cole et al. 2018 study also included an individual difference analysis revealing a positive relationship between preparation time and accuracy that was not susceptible to the within-subject "global fluctuations in performance" confound. (Or if the authors think that individual differences are susceptible to that confound it would be good to include a discussion of why.)

Response: This is a very good point. The reviewer is absolutely right that it is important to distinguish between the within-subject effects reported by Cole et al. (2018) (which might be confounded by global fluctuations – see our reply to Comment 3 of Reviewer 1) and individual differences effects they reported (which are indeed not confounded by global fluctuations). Accordingly, we refined the description of the Cole et al. (2018) study. Specifically, on p. 85, we now write: "Furthermore, two regression analyses revealed that the link between longer self-paced preparation and higher accuracy was stronger for novel tasks than for practiced tasks. First, using a single-trial regression, the authors found that participants had higher ACC in novel tasks when they prepared for a longer period compared to when they prepared for a shorter period. For practiced tasks, the authors observed a non-significant negative relationship (i.e., ACC numerically dropped when preparation time increased). Second, an individual-differences analysis (using the data of Experiment 1 and 3) showed that participants who prepared the task instructions longer were more accurate; this effect was stronger for novel tasks than for practiced tasks. Combined, these two analyses suggest a positive link between self-paced preparation and task performance."

In addition, we also clarified on p. 87 that we were only interested in within-subject effects here (given our focus on the benefit-cost analysis): "As illustrated in more detail in Appendix 1, this pair-split analysis is essentially a median split that controls for global fluctuations, and is therefore preferred over the single-trial regression approach used by Cole et al. (2018). (Note that the individual-differences regression approach used by Cole et al. is not confounded by global fluctuations either; however, we were exclusively interested in within-subjects effects in the present study, which is why we used our pair-split analysis.)"

3. It was noted that the Cole et al. 2018 study included a well-practiced task control condition, which did not show the positive prepRT-ACC relationship. It should be discussed whether this acts as a proper control for the within-subject "global fluctuations in performance" confound mentioned in the present study. It appears that it might be a proper control, given that the well-practiced tasks were intermixed with the novel tasks (such that global performance improvements over the session should in theory apply to both). Such a discussion would be especially important given that it could potentially resolve the discrepancy emphasized throughout the paper: that preparation time has tended to have a negative association with performance of well-practiced tasks (in contrast to a positive relationship for novel tasks). It is unclear why such a control condition was not included in any of the 4 experiments here (especially given the emphasis on the discrepancy in the literature), unless there were some concerns by the authors that such a control condition was not appropriate for some reason.

Response: We agree with the reviewer that practiced tasks could indeed be an appropriate control condition for novel tasks, and presenting both task types in an intermixed fashion like Cole et al. (2018) would allow one to control for global fluctuations. In the present study, however, we were less interested in the differences between self-paced preparation of novel tasks vs. practiced tasks. Instead, our main focus was on benefits and costs of self-paced preparation of novel tasks specifically, which is why we did not include practiced tasks as control condition (as this would double the number of conditions again). We acknowledged both issues in the General Discussion on p. 109: "Note, however, that a direct comparison between the present study and the work of Meiran et al. (2002) is somewhat difficult, since the present study does not contain any practiced tasks. The reason is that we were primarily interested in investigating potential benefits and costs of self-paced preparation for novel tasks specifically (and adding practiced tasks would have doubled the duration of the experiment, which was not feasible for practical reasons). Nevertheless, it is important to discuss the role

of self-paced preparation for performance in novel and practiced tasks.”(p. 109) and on p. 110: “For a better understanding of the link between self-paced preparation and performance in both novel and practiced tasks, future studies should take two aspects into account. First, and similar to Cole et al. (2018), the studies could present both novel and practiced tasks in an intermixed fashion, so that potential effects of global fluctuations should in theory apply to both task types. Second, the studies could also explicitly control for global fluctuations, for example by using the pair-split analysis that we introduced in the present study.”(p. 110).